# Programmable viscoelasticity in protein-RNA condensates with disordered sticker-spacer polypeptides

Ibraheem Alshareedah [1], Mahdi Muhammad Moosa [1], Matthew Pham[2], Davit A. Potoyan [2✉] & Priya R. Banerjee [1✉]

Liquid-liquid phase separation of multivalent proteins and RNAs drives the formation of biomolecular condensates that facilitate membrane-free compartmentalization of subcellular processes. With recent advances, it is becoming increasingly clear that biomolecular condensates are network fluids with time-dependent material properties. Here, employing microrheology with optical tweezers, we reveal molecular determinants that govern the viscoelastic behavior of condensates formed by multivalent Arg/Gly-rich sticker-spacer polypeptides and RNA. These condensates behave as Maxwell fluids with an elastically-dominant rheological response at shorter timescales and a liquid-like behavior at longer timescales. The viscous and elastic regimes of these condensates can be tuned by the polypeptide and RNA sequences as well as their mixture compositions. Our results establish a quantitative link between the sequence- and structure-encoded biomolecular interactions at the microscopic scale and the rheological properties of the resulting condensates at the mesoscale, enabling a route to systematically probe and rationally engineer biomolecular condensates with programmable mechanics.

[1] Department of Physics, University at Buffalo, Buffalo, NY 14260, USA. [2] Department of Chemistry, Iowa State University, Ames, IA 50011, USA. ✉email: potoyan@iastate.edu; prbanerj@buffalo.edu

Biomolecular condensates represent a class of dynamic membrane-less bodies that are central in compartmentalizing subcellular biochemical processes in viruses[1], bacteria[2,3], yeast[4,5], and human[6]. Past studies have reported a variety of biomolecular condensates with a broad range of physiological functions including stress response[7,8], mRNA processing[9], transcriptional activity control[10–12], and genome organization[13]. The dynamic liquid-like condensates formed by RNA-binding proteins, such as FUS, TDP43, and hnRNPA1 can further undergo a liquid-to-solid transition over time (known as maturation or aging of condensates), which can lead to pathological aggregates[7,14–16]. It is now generally accepted that the material states and dynamical properties of the protein condensates, which include viscosity, surface tension, network elasticity, and transport properties of constituent macromolecules, are key determinants of their biological functions and pathological effects inside living cells[6,17,18]. Typically, dynamic liquid-like micro-environments are deemed ideal for active regulation of biomolecular reactions in a signaling event, whereas irreversible condensate maturation has been linked to neurodegenerative disorders as well as tumorigenesis[19,20]. However, identifying the physiochemical factors that determine the material properties of biomolecular condensates remains a key challenge due to the complex dynamical properties of condensate fluid network across different length and timescales.

A central force that drives the formation of biomolecular condensates is multivalent interactions between protein and RNA chains, leading to a liquid-liquid phase separation and/or a percolation transition[21–23]. Multivalency in a protein chain is typically encoded either by intrinsically disordered regions (IDRs) with amino acid repeat sequences (such as the Arg−Gly−Gly or RGG repeats), by folded domains (such as $SH_3$ and PRM modules), or a combination thereof[22]. Multivalent homotypic protein-protein and heterotypic protein-RNA interactions driving biomolecular condensation have been described by the stickers-and-spacers polymer framework[22,24–26]. For IDRs, the residues that can enable inter-chain attractive interactions include arginine (R) in R/G-rich IDRs and tyrosine (Y) in prion-like IDRs[27] and are usually referred to as stickers. Additionally, the linker residues connecting these stickers are considered as spacers. The patterning of stickers and spacers can alter the physical properties of condensates and their phase behavior[27,28]. In theory[29], it is conceivable that network fluidity and stiffness can be encoded by the sequence composition and sticker identity in a polypeptide chain. Hence, understanding how the sticker-spacer architecture of disordered protein chains regulates the physical properties of biomolecular condensates will be insightful for understanding biological mechanisms and designing synthetic condensates.

Several recent studies have indicated that biomolecular condensates are network fluids with variable viscoelastic properties[21,22,30–33]. The viscoelasticity is presumably a result of transient network-like structures that form via physical cross-linking among protein and/or RNA chains with finite bond lifetime[22,34]. This has led to a growing interest in utilizing suitable experimental methods to probe condensate material properties across different timescales. In this study, we adopt passive microrheology with optical tweezers (pMOT) to quantify the viscoelastic properties of a series of artificial condensates formed by disordered sticker-spacer polypeptides and RNA. We find that at shorter timescales, peptide-RNA condensates have an elastically dominant rheological response, while at longer timescales, the same condensates behave as predominantly viscous liquids. The network relaxation time, viz., the timescale at which the condensate transitions from an elastically dominant behavior to a viscous behavior, is determined by the chemical identities of the sticker and spacer residues in the polypeptide chain. Using complementary biophysical assays and molecular dynamics simulations, we show that the variable viscoelastic behavior of condensates across different polypeptide variants is strongly correlated with differences in the strength of inter-chain attractions. Accordingly, we stipulate generalizable sequence heuristics that govern the viscoelasticity of peptide-RNA condensates and connect the same to their thermodynamic phase behavior. Utilizing this acquired knowledge of sequence-phase behavior-material property relations, we test simple strategies to fine-tune the condensate viscoelasticity over multiple orders of magnitude. Similar to the polypeptides, we further show that RNA sequence and shape influence the mechanics of peptide-RNA condensates. Our findings shed light on the origin of viscoelasticity in biomolecular condensates and allude to its interconnection with fundamentally relevant physical properties such as inter-molecular attractive interactions and temperature-dependent phase behavior.

## Results

**Passive microrheology with optical tweezers (pMOT) offers a quantitative method for characterizing the viscoelastic properties of biomolecular condensates.** Establishing a molecular grammar of sequence-encoded protein-protein and protein-RNA interactions that governs the biomolecular condensate dynamical properties requires systematic and high-resolution measurements of the condensate viscoelastic properties. Direct recording of the frequency-dependent viscous and elastic moduli can provide such insights[22]. The linear viscoelastic (LVE) behavior of homotypic protein condensates has recently been studied by active oscillatory microrheology using a dual-trap optical tweezer[30,31]. However, passive non-oscillatory measurements using a single optical trap offers an attractive orthogonal route to probe the LVE properties of complex fluids due to a wide range of experimentally accessible frequencies[35,36]. Furthermore, passive microrheology with optical tweezers (pMOT) only requires a single optical trap and does not need independent calibration measurements[37–40], making it convenient and well-suited for relatively high throughput studies. Here, we employ in-droplet pMOT to map the sequence-encoded and frequency-dependent viscoelastic properties of polypeptide-RNA condensates in vitro over three decades of frequencies from a single measurement (Supplementary Note 1). Briefly, 1 μm polystyrene beads are passively embedded within a condensate (placed on a glass surface) with each condensate containing at least one probe particle (Fig. 1a). The bead is constrained within the condensate by an optical trap (Fig. 1b). The motion of the bead inside the condensate is driven by the thermal fluctuations of the medium (the condensate) and constrained by the harmonic potential of the optical trap (Fig. 1c–e). The complex modulus of the condensate can be calculated from the normalized position autocorrelation function [NPAF, $A(\tau)$] of the bead (Fig. 1f) and the trap stiffness $\kappa$ as[37,39]

$$G^*(\omega) = \frac{\kappa}{6\pi a}\left(\frac{i\omega\hat{A}(\omega)}{1 - i\omega\hat{A}(\omega)}\right) = G'(\omega) + iG''(\omega) \qquad (1)$$

Where $a$ is the bead radius and $\hat{A}(\omega)$ is the Fourier transform of $A(\tau)$ (see Supplementary Note 1 for further details). In Eq. (1), $G'$ and $G''$ represent the elastic (storage) and viscous (loss) modulus of the condensate, respectively. Using pMOT, we first measured the frequency-dependent viscoelastic moduli of a model peptide-RNA condensate formed by a short disordered lysine-rich repeat polypeptide $[KGKGG]_5$ and a homopolymeric RNA $[rU]_{40}$ (rU40) (Fig. 1a). We found that $[KGKGG]_5$-rU40 condensates display a complex modulus that is dominated by the viscous component ($G''$) at both low and high frequencies (1 to 100 Hz),

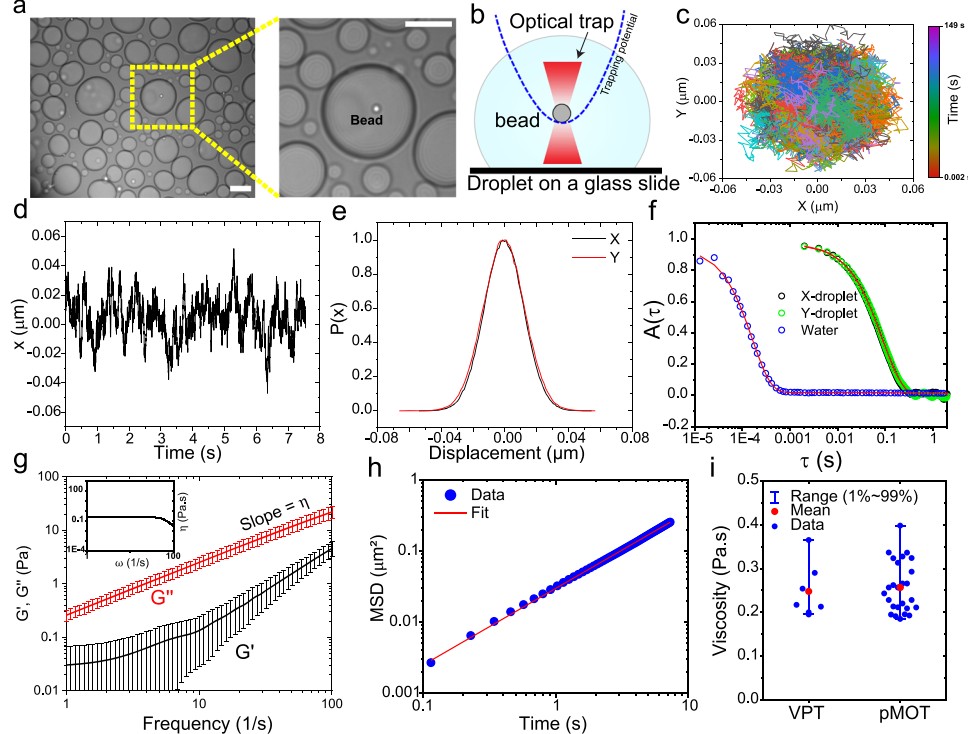

**Fig. 1 Determination of frequency-dependent viscoelastic moduli of peptide-RNA condensates using passive microrheology with optical tweezers (pMOT). a** A bright-field image showing a polystyrene bead (1 µm) trapped within a [KGKGG]₅-rU40 condensate using an optical trap. Scale bar = 10 µm. **b** A conceptual scheme of the pMOT experiment. The bead is optically trapped within a biomolecular condensate sitting on a microscope glass surface. **c** A representative 2D trajectory of the bead shown in (a) within the optical trap inside a [KGKGG]₅-rU40 condensate. **d** The trajectory of the trapped bead in the X-direction. **e** Normalized distribution of displacements along the X- and Y-directions for the trajectory in c and d. **f** The normalized position autocorrelation function [NPAF, A(t)] as calculated from the trajectory in **c** for a bead that is optically trapped inside [KGKGG]₅-rU40 condensate (green and black) and inside water (blue) as a reference. Solid lines are multi-exponential fits (see Supplementary Note 1). **g** The average viscoelastic moduli as obtained from normalized position autocorrelation function using equation-1 for [KGKGG]₅-rU40 condensates. G′ and G″ represent the elastic and viscous modulus, respectively. Solid lines are averages of the moduli of 10-20 condensates. Error bars represent the standard deviation as calculated from the moduli of 10-20 condensates. *Inset*: frequency-dependent condensate viscosity as determined from the viscous modulus using the relation $\eta(\omega) = G''(\omega)/\omega$. **h** The ensemble-averaged mean square displacement (MSD) of 200 nm polystyrene beads within [KGKGG]₅-rU40 condensates using video particle tracking (VPT) microrheology in absence of optical traps (see Methods section for further details). **i** Comparison between the zero-shear viscosity as determined by pMOT (n = 26 measurements over 3 independent samples) and VPT-derived (n = 7 measurements over 3 independent samples) viscosity. Error bars represent the range of the data.

indicating a predominant liquid-like behavior (Fig. 1f, g). The zero-shear viscosity ($\eta$) of [KGKGG]₅-rU40 condensates, which is directly obtained from the slope of the viscous modulus ($\eta = G'/\omega$), is ~0.26 ± 0.06 Pa.s. This is ~300-fold higher than the viscosity of water and is nearly independent of the frequency (Fig. 1g), suggesting that these condensates behave mostly as a viscous liquid throughout the entire experimentally accessible frequency range. To cross-validate our results obtained from pMOT, we independently measured the viscosity of [KGKGG]₅-rU40 condensates using video particle tracking (VPT) microrheology[41] and found good agreement (Fig. 1h, i). Therefore, pMOT presents a suitable method to probe the frequency-dependent LVE properties of biomolecular condensates in vitro (see Supplementary Note 1).

**Tunable LVE properties of peptide-RNA condensates formed by sticker-spacer polypeptides: A. Role of "sticker" residues.** To probe how polypeptide sequence features govern the condensate viscoelastic properties, we utilized short modular multivalent polypeptides that undergo phase separation with RNAs[41–50]. The design of these peptide sequences was inspired by naturally occurring Arginine/Glycine-rich (R/G-rich) IDRs in eukaryotic ribonucleoproteins (RNPs)[51,52], which function as promiscuous

RNA-binding motifs. Previous studies by our group and others have reported that Arginine in R/G-rich IDRs can be classified as stickers that potentiate RNP phase separation with RNA through a combination of electrostatic, cation-π and π-π interactions[44,48,53]. Due to their multivalent architecture, we hypothesized that the R/G-rich IDRs have the potential to serve as programmable stickers-and-spacers polypeptides with tunable condensation behavior in presence of RNA (Fig. 2a). To test this, we designed a synthetic repeat polypeptide encompassing 5 repeats of the RGRGG motif, [RGRGG]₅. pMOT experiments on [RGRGG]₅-rU40 condensates revealed dominant elastic behavior at high frequencies (short timescales) and viscous behavior at low frequencies (long timescales). This frequency-dependent mechanical behavior of R/G-repeat condensates is reminiscent of a typical Maxwell fluid[37] (Fig. 2b) with a single crossover frequency between the elastically dominant and predominantly viscous regimes. This crossover frequency is the inverse of the terminal relaxation time ($\tau_M$) of the condensate network (Fig. 2b), which signifies the average peptide-RNA network reconfiguration time (or bond lifetime)[35]. For [RGRGG]₅-rU40 condensates, $\tau_M$ is 60 ± 10 ms, meaning that in timescales shorter than 60 ms, the [RGRGG]₅-rU40 condensates behave as an elastic solid (with G′ > G″) and vice-versa (Fig. 2b). This is in

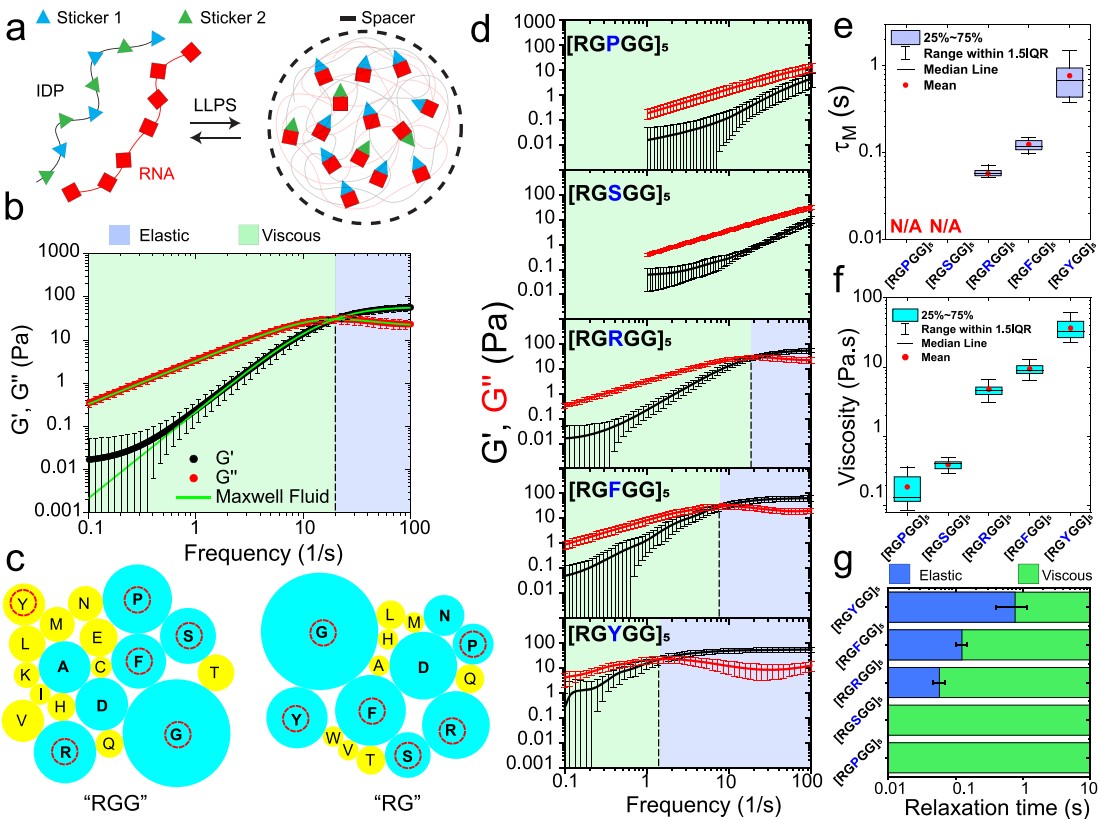

**Fig. 2 Sequence-dependent control over linear viscoelastic (LVE) behavior of peptide-RNA condensates. a** A scheme showing the sticker-spacer architecture of associative peptide and RNA chains. Here, sticker-RNA interactions drive the condensation. **b** A plot showing the average elastic modulus (G', black) and the average viscous modulus (G", red) of [RGRGG]₅-rU40 condensates ($n = 17$ measurements over 3 independent samples). Green lines are fits to experimental data using a single-mode Maxwell fluid model. The crossover frequency is indicated by the black dashed line and is the inverse of terminal relaxation time $\tau_M$. Shaded regions represent the dominant elastic regime (light-blue) and the dominant viscous regime (light-green), respectively. Error bars represent one standard deviation (±1 s.d.). **c** The relative abundance of different amino acids within the inter-RG/RGG spacers of RG/RGG motifs in human RNA-binding proteins represented as bubble charts. Sizes of individual bubbles represent the fraction of inter-RG/RGG spacers that contain the corresponding amino acid (see Methods section). Residues that occurred in more than 10% of the analyzed protein sequences (RG/RGG domains from 407 human RNA-binding protein sequences) are indicated in cyan. Residues marked with red circles are utilized for peptide design for the experimental studies reported here. **d** The average frequency-dependent viscoelastic moduli of [RGXGG]₅-rU40 condensates where X = P/S/R/F/Y. The crossover frequencies are indicated by black dashed lines. Shaded regions represent the dominant elastic regime (light-blue) and the dominant viscous regime (light-green), respectively. Error bars represent ±1 s.d. **e** The terminal relaxation time $\tau_M$ of [RGXGG]₅-rU40 condensates as measured by pMOT (X = P/S/R/F/Y). **f** The zero-shear viscosity of [RGXGG]₅-rU40 condensates as measured by pMOT (X = P/S/R/F/Y). **g** An LVE state diagram indicating the timescales at which the elastic modulus dominates (blue) and the timescales where the viscous modulus dominates (green) in [RGXGG]₅-rU40 condensates (X = P/S/R/F/Y). Error bars represent one standard deviation (±1 s.d.). For **d–g**, the sample size is $n = (29,16,17,17,16)$ measurements over 3 independent samples for X = (P,S,R,F,Y), respectively.

stark contrast with [KGKGG]₅-rU40 condensates which behave as a predominantly viscous liquid (Fig. 1g) within the same frequency range. Consistent with this observation, the zero-shear viscosity of R/G-rU40 condensates is ~5 ± 1 Pa.s (Supplementary Fig. 1), which is ~20-fold higher than the same for K/G-rU40 condensates. This difference in viscosity between R/G and K/G condensates is consistent with previously published reports[28,44,45,48] and signifies that the viscoelastic properties of these condensates can vary based on the choice of stickers (i.e., Arg vs. Lys).

Besides Arg and Gly residues, R/G-rich domains in natural RNPs frequently contain aromatic and polar residues[49]. Our bioinformatics analysis of RG/RGG motifs from >400 human RNA-binding proteins[54] reveal that R/G-rich IDRs are primarily interspersed with four uncharged amino acids: Tyrosine (Y), Phenylalanine (F), Serine (S), and/or Proline (P) (Fig. 2c). The occurrence of these four residues within R/G-rich IDRs can be important for modulating self-interactions and interactions with

RNA and/or solvent-mediated interactions. As such, these sequence variations may control the material properties of polypeptide-RNA condensates. To test this idea systematically, we designed a variable sticker-spacer polypeptide sequence, [RGXGG]₅, where the amino-acid X can be Y, F, S, P, or simply R (Supplementary Table 1). In the context of our work, amino acids, such as Arg or Lys that primarily contribute to the RNA binding are defined as stickers, while spacers are linker residues connecting the stickers, such as Gly. Employing our pMOT experiments on condensates formed by [RGXGG]₅ and rU40 RNA (Supplementary Fig. 2), we found a rich variation in the viscoelastic behavior that spans two orders of magnitude depending on the identity of the amino acid X. In discussing these results, we choose [RGRGG]₅ as our reference system (X = Arg). Setting X = Phe, we observed a 2-fold increase in the terminal relaxation time from 60 ± 10 ms to 120 ± 20 ms, and ~2-fold increase in the zero-shear viscosity of peptide-rU40 condensates (Fig. 2d–f). Intriguingly, setting X = Tyr resulted

in condensates that exhibited an order of magnitude increase in both the terminal relaxation time (~800 ± 400 ms; Fig. 2d, e) and the zero-shear viscosity (~40 ± 10 Pa.s; Fig. 2e, f & Supplementary Fig. 3). The high terminal relaxation time for $[RGYGG]_5$-rU40 condensates ($\tau_M \sim 0.8$ s) indicates that these condensates are dominantly elastic over a much longer timescale than $[RGRGG]_5$-rU40 and $[RGFGG]_5$-rU40 condensates (Fig. 2g). We note that, in the context of known biological assemblies, the observed LVE behavior of $[RGYGG]_5$-rU40 condensate is comparable to that of the reconstituted extracellular matrix[55]. Consistent with these results, video particle tracking microrheology using free poly-styrene beads[41] (without optical traps) revealed two clear diffusion modes for $[RGYGG]_5$-rU40 condensates, representing distinguishable elastic and viscous regimes (Supplementary Fig. 4). The first mode at shorter timescales displayed a near-constant MSD as a function of time, whereas a transition to a linear diffusion ($MSD = 4D\tau^\alpha$; $\alpha \sim 1$) was observed at ~1200 ms. This is comparable to the terminal relaxation time obtained from pMOT (800 ± 400 ms; Supplementary Fig. 4). At the same time, the MSD obtained from $[RGRGG]_5$-rU40 condensates did not show a flat regime above 100 ms (the shortest lag time in the MSD for this sample), indicating mostly viscous behavior. This is consistent with results obtained from the pMOT assay (Supplementary Fig. 4) since the terminal relaxation time of $[RGRGG]_5$-rU40 condensates is ~60 ms, beyond which the viscous modulus dominates the rheological behavior (Fig. 2e).

Contrary to X = Tyr or Phe, setting X = Pro or Ser resulted in a complete loss of elasticity across the tested frequencies (Fig. 2d) akin to the $[KGKGG]_5$-rU40 condensates. The zero-shear viscosity of $[RGPGG]_5$-rU40 condensates was ~0.19 ± 0.09 Pa.s, which is ~25-fold lower than the $[RGRGG]_5$-rU40 condensates (Fig. 2f) and ~200 fold lower than the $[RGYGG]_5$-rU40 condensates. Similarly, the viscosity of $[RGSGG]_5$-rU40 condensates was found to be 0.40 ± 0.06 Pa.s. Taken together, our rheological measurements reveal that tyrosine, phenylalanine, and arginine are promoters of elasticity in R/G-repeat polypeptide-RNA condensates, while lysine, proline, and serine residues promote the formation of predominantly viscous condensates (Fig. 2g; Supplementary Fig. 3; Supplementary Table 2).

We next asked whether the observed alterations in the LVE properties of peptide-RNA condensates are reflected in biomolecular diffusion within condensates. To assess that, we measured molecular mobility of a fluorescently-labeled RNA client, rU10 using fluorescence recovery after photobleaching (FRAP). We found that the rate of fluorescence recovery of the client RNA within $[RGPGG]_5$-rU40 condensates ($\tau = 4$ s) was significantly higher than that of $[RGRGG]_5$-rU40 ($\tau = 23$ s), whereas $[RGYGG]_5$-rU40 condensates displayed lowest recovery rate ($\tau = 74$ s; Supplementary Fig. 5). The rank order of RNA mobility therefore follows the same order as the dynamical properties of these condensates, providing further evidence that these condensates have tunable material properties as the stickers in the polypeptide chain are varied. To further characterize the physical properties of condensates as a function of polypeptide repeat sequence, we utilized a variable-size dextran recruitment assay[56,57] and measured the average mesh size of the condensates with the weakest and strongest viscoelastic behavior (i.e., $[RGPGG]_5$-rU40 and $[RGYGG]_5$-rU40 condensates, respectively). We observed that the apparent mesh size of $[RGPGG]_5$-rU40 is ~6 nm, which is 50% larger than that of $[RGYGG]_5$-rU40 (~4 nm) (Supplementary Figs. 6 and 7), showing a correlation with the condensate viscoelastic properties. Collectively, our results reveal that the structure and material properties of peptide-RNA condensates are governed by the choice of RGXGG motifs in our designed polypeptide sequences.

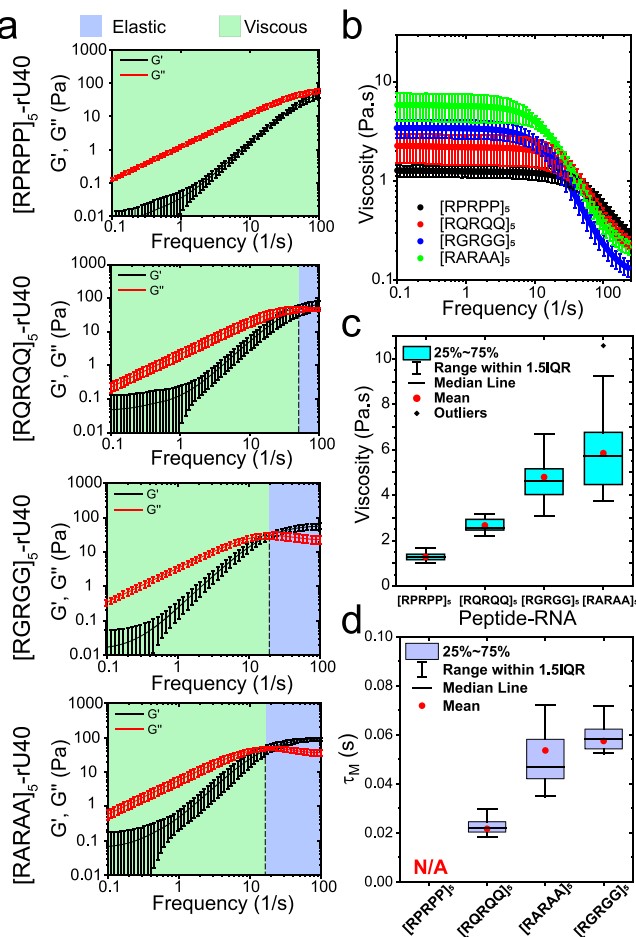

**Fig. 3 Effect of spacer sequence variation on the frequency-dependent viscoelasticity of peptide-RNA condensates. a** Average viscoelastic moduli of $[RxRxx]_5$-rU40 condensates. The crossover frequencies are indicated by black dashed lines. Shaded regions represent the dominant elastic regime (light-blue) and the dominant viscous regime (light-green), respectively. Error bars represent one standard deviation (±1 s.d.). **b** The frequency-dependent viscosity of $[RxRxx]_5$-rU40 condensates as measured by pMOT. Error bars represent one standard deviation (±1 s.d.). **c** The zero-shear viscosity of $[RxRxx]_5$-rU40 condensates as measured by pMOT. **d** The terminal relaxation time $\tau_M$ of $[RxRxx]_5$-rU40 condensates as measured by pMOT. For the data shown in **a**–**d**: $x = P/Q/G/A$. For the data in **a**–**d**, the sample size is $n = (18,14,18,17)$ measurements for $x = (P,Q,A,G)$, respectively.

**Tunable LVE properties of peptide-RNA condensates formed by sticker-spacer polypeptides: B. Role of "spacer" residues.** In addition to stickers, spacer residues may also impact the condensate dynamical properties by altering the RNA-binding affinity and/or modulating the effective solvation volume of the polypeptide chain[26]. To explore such an effect, we designed a generic poly-peptide $[RxRxx]_5$ where x represents either glycine, glutamine, alanine, proline, or leucine (G, Q, A, P, or L, respectively). We observed that G-to-P substitutions substantially reduced the condensate viscosity (Fig. 3a–c) and resulted in ~6-fold lower relaxation time ($\tau_M \sim 10$ ms since the crossover is slightly beyond the experimental frequency range; Fig. 3a). However, as compared to $[RGPGG]_5$ ($\eta \sim 0.19 \pm 0.09$ Pa.s, Fig. 2f), $[RPRPP]_5$-rU40 condensates showed a higher zero-shear viscosity (~1.3 ± 0.2 Pa.s). This indicates that replacing arginine (i.e., stickers) residues with proline directly impacts the peptide-RNA interactions and results in larger effects on the condensate rheological properties. For RQRQQ

repeats, we also found a reduced viscosity and relatively weaker elastic component than [RGRGG]$_5$-rU40 condensates (Tables S1 and S2; Fig. 3a–c). In contrast, [RARAA]$_5$-rU40 condensates showed a comparable elastic response ($\tau_M = 50 \pm 20$ ms) and marginally higher zero-shear viscosity (~6 ± 2 Pa.s) than [RGRGG]$_5$-rU40 condensates (Fig. 3). Finally, mutating G-to-L ([RLRLL]$_5$) resulted in the formation of arrested networks instead of liquid-like condensates (Supplementary Fig. 8) under identical experimental conditions. These results highlight that the LVE properties of peptide-RNA condensates can independently be controlled via spacer residue variation (Fig. 3), and in combination with the stickers (Fig. 2), these alterations can further attenuate or magnify condensates' viscoelastic behavior.

**The LVE properties of peptide-RNA condensates are correlated with the strength of attractive interactions between polypeptide and RNA chains.** Our findings discussed above reveal sequence heuristics that encode viscoelasticity in peptide-RNA condensates. By altering the chemical identity of the stickers and spacers in disordered repeat polypeptides of the same length and similar patterning, the dynamical properties of these condensates could be varied by almost

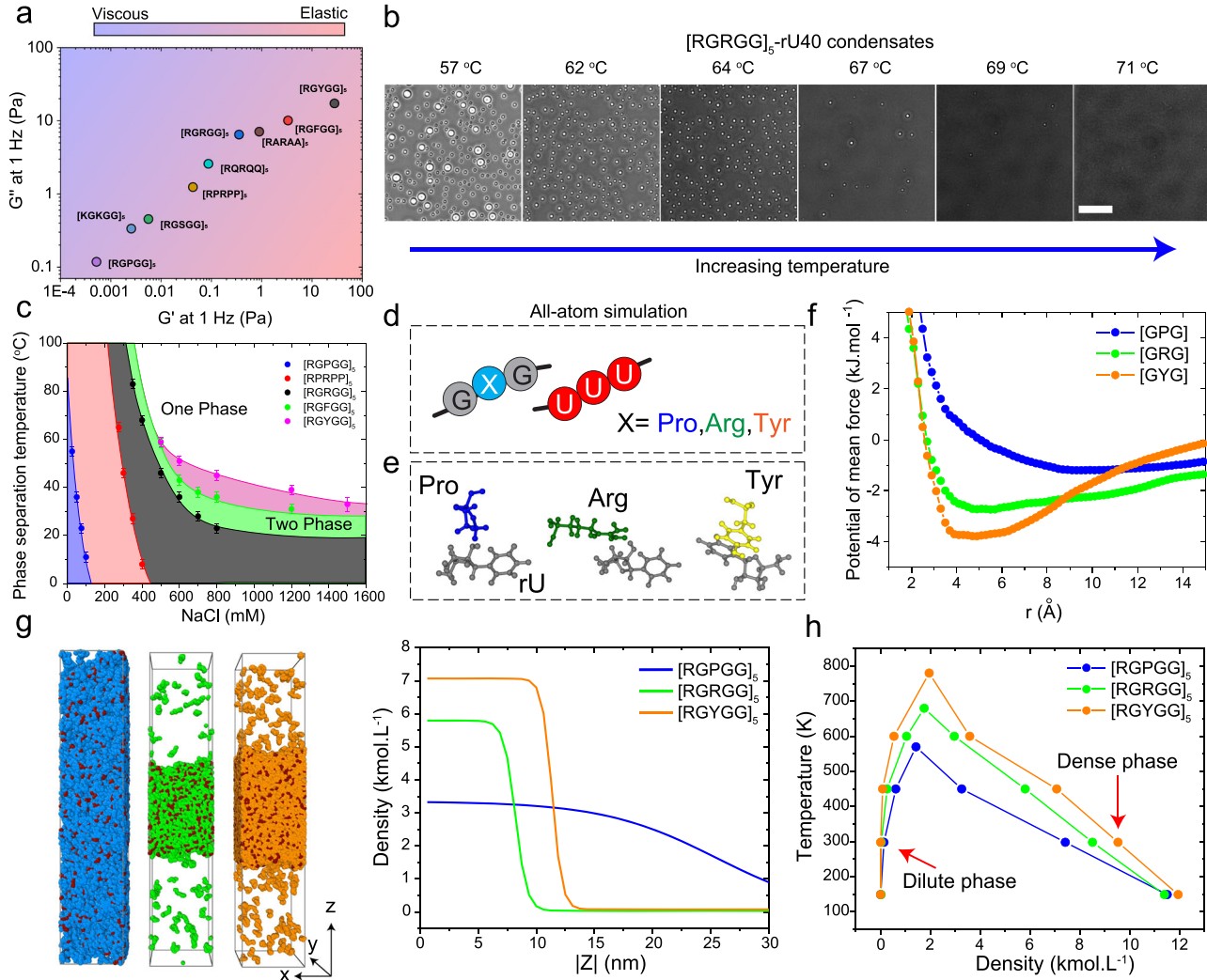

**Fig. 4 Peptide-RNA interactions govern the condensate LVE properties. a** A material state diagram showing the elastic (G′) and viscous (G″) moduli at 1 Hz frequency for the various condensates tested in this study. This plot shows a broad range of tunability in condensate viscoelasticity via sequence perturbations. **b** Bright-field images of [RGRGG]$_5$-rU40 condensates showing phase separation upon cooling. Scale bar is 20 μm. **c** Temperature-salt state diagram for [RGPGG]$_5$, [RPRPP]$_5$, [RGRGG]$_5$, [RGFGG]$_5$, and [RGYGG]$_5$ mixtures with rU40 RNA. Shaded regions indicate the salt and temperature conditions that allow phase separation. The most stable peptide-RNA condensates are [RGYGG]$_5$-rU40 and the least stable are [RGPGG]$_5$-rU40 condensates. Error bars represent the temperature range between a phase separated sample and a homogeneous (not phase separated) sample. **d** A scheme summarizing the all-atom simulations of tri-amino acids [GXG] and a tri-nucleotide [rUrUrU] where X was set to either Arg or Pro or Tyr. **e** All-atom simulation equilibrium snapshots of Pro-rU, Arg-rU, and Tyr-rU interactions. Note the π-π stacking in the case of Tyr-rU pair that leads to stronger interactions with RNA. **f** Free energy profiles of [GXG]-[rUrUrU] attraction from model peptide-RNA all-atom constant temperature simulations shown in **e**. X is set to Arg, Pro, or Tyr. The tripeptide that is stickiest to [rUrUrU] is [GYG] while [GPG] has the weakest interaction with the trinucleotide. **g** Snapshots from the coexistence simulations with coarse grained molecular models of full peptide sequences and rU40 RNA showing the equilibrium configuration of peptide and RNA chains in [RGPGG]$_5$-rU40, [RGRGG]$_5$-rU40, and [RGYGG]$_5$-rU40 mixtures at a temperature of 450 K. The rU units are colored in red. The Pro units are colored in blue. The Arg units are colored in green. The Tyr units are colored in orange. Corresponding density profiles are shown along the z-direction of the box. **h** Temperature-density phase diagrams for [RGPGG]$_5$-rU40, [RGRGG]$_5$-rU40, and [RGYGG]$_5$-rU40 condensates as extracted from the coexistence simulations in **g**. At each temperature, the densities of the condensed and dilute phases are shown.

two orders of magnitude (Fig. 4a). To provide a thermodynamic basis for the observed differences in the condensate LVE properties, we consider the type and relative strength of interactions between polypeptide chains and the RNA. We and others have previously shown that the material properties of peptide-RNA condensates are governed by a combination of long-range interactions (such as attractive and/or repulsive Coulomb interactions) and short-range attractions (such as cation-π and π-π interactions)[44,48]. For cationic IDRs, arginine-rich polypeptides have higher affinity for RNA than the corresponding lysine variants due to the unique planar geometry and electronic structure of the arginine guanidino group, which allows for stronger cation-π interactions[44,48,58]. Such differences in protein-RNA interactions are reflected in the phase behavior of the mixture, revealing a significantly larger window of the phase-separated regime as a function of mixture composition for the arginine-rich polypeptide as compared to the corresponding lysine variant[44]. Therefore, the net polypeptide-RNA interaction strength is not only a critical determinant of the phase behavior, but it also regulates the dynamical properties of the condensate network. Accordingly, we hypothesized that the peptide-RNA combinations with the most pronounced condensate viscoelastic properties are those that form the most stable condensates. To test this idea, we looked at the stability of peptide-RNA condensates against two physical parameters that can control the strength of interactions in these systems: buffer ionic strength and temperature. Increasing salt concentration decreases the electrostatic attraction between peptide and RNA chains due to the electrostatic screening by counterions. Therefore, the condensation of negatively charged RNA with positively charged peptides (such as [RGRGG]$_5$) is usually suppressed at higher salt concentrations[41–44] (Supplementary Fig. 9a). The condensation behavior with temperature is rather more complex as biopolymers in solutions can undergo phase separation with an upper critical solution temperature (UCST), a lower critical solution temperature (LCST), or a combination thereof[59,60]. In our peptide-RNA system, we observed an UCST behavior (Fig. 4b). This UCST behavior suggests that inter-chain and/or inter-complex attraction drives phase separation (enthalpy-driven LLPS), and therefore, higher attractive interactions are expected to lead to a higher UCST[59,61–63]. Therefore, measuring thermo-responsive phase behavior can directly report on the relative strength of inter-molecular interactions across our various peptide-RNA mixtures.

To map the stability of condensates across a broad salt and temperature range, we used a combination of solution turbidity measurements and temperature-controlled bright-field microscopy (Fig. 4c). At room temperature (22 ± 1 °C), we find that [RGRGG]$_5$-rU40 condensates remain stable up to 800 mM NaCl. For [RGPGG]$_5$-rU40 condensates, which are significantly less viscoelastic than [RGRGG]$_5$-rU40, the critical salt concentration was found to be approximately 50 mM (Supplementary Fig. 9a). Intriguingly, [RGYGG]$_5$-rU40 condensates, which have stronger viscoelastic response than [RGRGG]$_5$-rU40 condensates, did not completely dissolve even at ~3000 mM NaCl (Supplementary Fig. 9a). These results suggest that the net attractive interactions driving the phase separation of peptide-RNA complexes are much stronger in the case of [RGYGG]$_5$-rU40 system than [RGRGG]$_5$ and [RGPGG]$_5$ complexes with RNA. Next, we measured the phase separation temperature (T$_{ph}$) of selected peptide-RNA mixtures with a broad range of dynamical properties at various salt concentrations and constructed a temperature-salt state diagram. We observed that [RGPGG]$_5$-rU40 condensates dissolve at a temperature of T$_{ph}$ = 55 ± 2 °C, while [RGRGG]$_5$-rU40 condensates have a T$_{ph}$ > 90 °C at the same experimental buffer containing 25 mM NaCl (Fig. 4c). Increasing salt concentration to 500 mM NaCl led to a lowering of T$_{ph}$ for [RGRGG]$_5$-rU40 condensates to 46 ± 2 °C (Fig. 4c). Under this condition, [RGPGG]$_5$-rU40 mixture did not phase separate at all the tested

temperatures (T$_{ph}$ < 5 °C). Importantly, T$_{ph}$ progressively increased from [RGRGG]$_5$ to [RGFGG]$_5$ to [RGYGG]$_5$ at a fixed salt concentration, thereby indicating increased strength of intermolecular interactions by P-to-R-to-F/Y substitutions (Fig. 4c). Similar to sticker variations, the substitution of G-to-P as spacers, which enhances the dynamicity of condensate network (Fig. 3), led to a significant lowering of USCT. Overall, our temperature-salt state diagram conveys two important findings, (i) The relative rank order of peptide-RNA condensate UCST provides a thermodynamic measure of net attractive interactions in the condensate network across different polypeptide variants, and (ii) the thermal and salt stability of peptide-RNA condensates and their linear viscoelastic properties are strongly correlated. In other words, peptide-RNA condensates with the most pronounced viscoelastic properties are those that are most stable against temperature and salt variation. Since condensate thermal stability is governed by the strength of intermolecular interactions that drive phase separation, we conclude that the strength of intermolecular interactions dictates the magnitude of viscoelasticity in peptide-RNA condensates.

To explore this observed correlation further, we performed all-atom molecular dynamics simulations probing the interactions between a generic oligomer GXG and a trinucleotide rU3 RNA (UUU; Fig. 4d). The variable amino acid X was set to Pro, Arg, or Tyr. We carried out 1 μs long explicit solvent simulations with tripeptide and tri-nucleotide units packed in a cubic solvated and ionized box at ~100 mg/ml density, temperature T = 300 K, salt concentration = 25 mM and pressure of 1 atmosphere using the amber 99SB-disp*-ildn force field with TIP3P model for water[64]. We extracted potentials of mean force of center of mass coordinates of residues (see Methods for simulation details). Our results show a clear and robust trend of Tyr (GYG) being the sticker residue outflanking the charged Arg (GRG) and uncharged Pro (GPG; Fig. 4e, f and Supplementary Fig. 9b) in their capacity to bind to nucleotides. This is consistent with our observation that Tyr-rich peptide [RGYGG]$_5$ forms condensates that have more dominant elastic behavior than Arg-rich peptide and Pro-rich peptides ([RGRGG]$_5$ and [RGPGG]$_5$). Upon close inspection of the free energy curves (Fig. 4f), we find that Tyr forms stronger short-range π-π ring contacts with Uracil bases compared to the electrostatic and salt bridge contacts, which leads to Tyr-U bonds being longer-lived than Arg-U and Pro-U contacts (Fig. 4e). We note that similar binding free energy trend (Tyr > Arg > Pro) has also been obtained through restrained simulations of single solvated amino acid and nucleobases under physiological salt conditions using different force field[65]. These differences in interaction strengths also explain the higher stability of Tyr-rich condensates against salt and temperature variation (Fig. 4c and Supplementary Fig. 9a). The microscopic insights obtained from our all-atom simulations were subsequently used to carry out liquid-vapor phase coexistence simulations with the full-length peptide and RNA sequences by using coarse-grained representation of chains with 1 bead per amino acid/nucleotide[42,66]. Here we calibrate the short-range attraction of residues Pro, Arg and Tyr towards RNA (Uracil) units by tuning the short-range energy coefficient of a standard Lennard-Jones potential to match the difference in the free energy minima from the all-atom simulations. Electrostatic forces are accounted for through the standard Debye-Huckel term. Inspecting the equilibrium configurations and plotting the density profiles across the simulation box, we found that [RGYGG]$_5$-rU40 showed the strongest phase separation behavior with the highest molecular density in the condensed phase (Fig. 4g). This was followed by the [RGRGG]$_5$-rU40 system which also showed strong phase separation albeit with a lesser density in the condensed phase than [RGYGG]$_5$-rU40 system. For [RGPGG]$_5$-rU40 mixture, phase

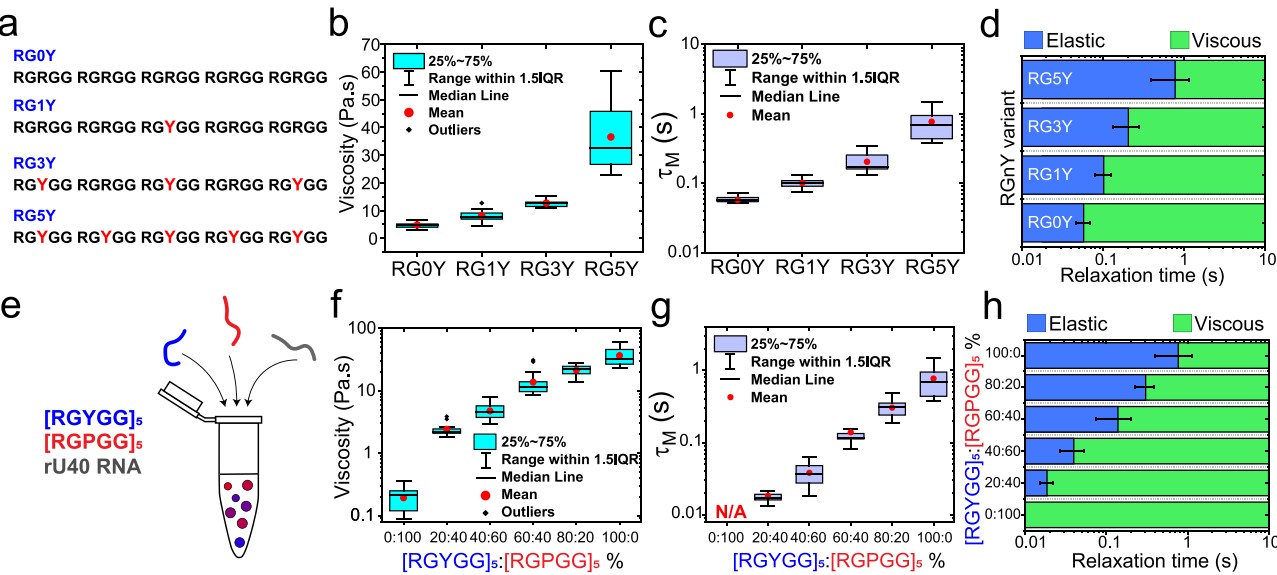

**Fig. 5 Continuous tuning of LVE behavior of peptide-RNA condensates by two orthogonal approaches. a** Polypeptide sequences of the RG*n*Y peptide design with *n* representing the number of Tyr residues. RG0Y corresponds to [RGRGG]₅, RG5Y corresponds to [RGYGG]₅ (see Supplementary Table 1). **b** The zero-shear viscosity of RG*n*Y-rU40 condensates as measured by pMOT. **c** The terminal relaxation time $\tau_M$ of RG*n*Y-rU40 condensates as measured by pMOT (see Supplementary Fig. 10). **d** An experimental LVE state diagram indicating the timescales at which the elastic modulus dominates (blue) and the timescales where the viscous modulus dominates (green) in RG*n*Y-rU40 condensates. Error bars represent one standard deviation (±1 s.d.). **e** A scheme showing the preparation of condensates formed by mixing [RGYGG]₅, [RGPGG]₅, and rU40 RNA with variable [RGYGG]₅-to-[RGPGG]₅ ratios. Here, the total peptide concentration is fixed at 5.0 mg/ml and the relative fractions of [RGYGG]₅ and [RGPGG]₅ are varied. **f** The zero-shear viscosity of condensates formed by mixtures of [RGYGG]₅, [RGPGG]₅, and RNA at variable [RGYGG]₅-to-[RGPGG]₅ ratios. **g** The terminal relaxation time ($\tau_M$) of the same condensates as in **f**. See Supplementary Fig. 12 for the viscoelastic moduli. **h** An experimental LVE state diagram indicating the timescales at which the elastic modulus dominates (blue) and the timescales where the viscous modulus dominates (green) in [RGYGG]₅-[RGPGG]₅-rU40 condensates as a function of [RGYGG]₅-to-[RGPGG]₅ ratio. Error bars represent one standard deviation (±1 s.d.). For **b–d**, the sample size is n = (17,20,10,16) for (RG0Y, RG1Y, RG3Y, RG5Y), respectively. For **g–h**, the sample size is n = (29,14,36,18,18,16) for [RGYGG]₅ fractions of (0%,20%,40%,60%,80%,100%), respectively.

separation was much less pronounced as evidenced by the blurred interface and a mild change in density across the condensed-dilute phase interface (Fig. 4g). These results show that by tuning intermolecular interactions, the molecular density of the condensed phase is altered. Next, we repeated the coexistence simulations at various temperature conditions and measured the molecular density in the condensed and dilute phases, which was subsequently plotted as a temperature-density phase diagram (Fig. 4h). We find that [RGYGG]₅-rU40 condensates show the highest stability against temperature followed by [RGRGG]₅-rU40 system and [RGPGG]₅-rU40 condensates. These results are in strong agreement with our experimental temperature-salt phase diagram (Fig. 4c). In summary, our computational analysis suggests that sequence-encoded peptide-RNA interactions lead to distinct molecular density of the condensed phase (Fig. 4g), the phase separation temperature of the system (Fig. 4h), and the bond lifetime within the condensate network (Fig. 4f), all of which are key determinants of the condensate material properties. These results also draw attention to the complex interplay of interactions (such as ionic, cation-π and π-π) that leads to emergent affinities between peptides and RNA beyond the monomer-monomer interactions that are often used to explain biophysical properties of biomolecular condensates.

**Strategies for fine-tuning the LVE properties of biomolecular condensates.** Intracellular biomolecular condensates are typically multicomponent and are known to adopt a diverse range of material states from viscous fluids to viscoelastic solids[21]. These observations have inspired our search to understand the physi-cochemical factors that control biomolecular condensates' viscoelastic behavior and devise molecular-level strategies to fine-tune the same. Based on our results on binary peptide-RNA condensates (Fig. 4a), we first considered two orthogonal strategies to form condensates with adjustable viscoelasticity − (a) by sequential perturbation of sticker identity in repeat polypeptides; and (b) by mixing multiple peptides with orthogonal LVE behavior. Based on our results showing enhanced viscoelasticity of peptide-rU40 condensates upon replacing all RGRGG motifs with RGYGG motif (Fig. 2 and Fig. 4a), we considered a sequential introduction of RGYGG motifs replacing the RGRGG motifs in the [RGRGG]₅ peptide (Fig. 5a and Supplementary Table 1). This sequence perturbation strategy led to a progressive enhancement of the viscosity (~5–40 Pa.s) and the terminal relaxation time (~60–800 ms) of peptide-rU40 condensates as a function of the number of R-to-Y substitutions (Fig. 5b–d and Supplementary Fig. 10). In our second set of experiments, instead of a single repeat polypeptide, we utilized a mixture of two polypeptides with orthogonal LVE behavior, viz., [RGYGG]₅, which forms predominantly elastic condensates, and [RGPGG]₅, which forms viscous droplets with rU40 RNA (Figs. 2d, 5e and Supplementary Table 2). We observed that [RGYGG]₅ acts as a dopant that progressively enhances the viscoelasticity of [RGPGG]₅ condensates with rU40 (Fig. 5f–h). The ternary peptide-RNA mixtures ([RGYGG]₅, [RGPGG]₅, and rU40) formed homogeneous condensates at all mixing ratios with LVE properties that can be controlled by the molar ratio of [RGYGG]₅ and [RGPGG]₅ (Supplementary Figs. 11 and 12). We observed a monotonic increase in the zero-shear viscosity (~0.19–40 Pa.s) and the terminal relaxation time (from <10 to 800 ms) as a

function of the [RGYGG]₅:[RGPGG]₅ ratio (Fig. 5f, g). These results suggest a plausible mechanism for the regulation of the LVE properties in multi-component biomolecular condensates, where the similarity in linear sequences promotes the formation of well-mixed condensates (as opposed to multiphasic condensates[67]) and the dissimilarity in sticker identity allows for composition-dependent regulation of the condensate viscoelasticity (Fig. 5h). Taken together, our presented results demonstrate simple design strategies to control the mechanical behavior of biomolecular condensates formed by stickers-and-spacers polypeptides (Fig. 5).

**Tunable LVE properties of peptide-RNA condensates by RNA sequence and structure.** So far in this study, our discussions have been focused on the role of polypeptide sequence variation on the condensate material properties. Evidences are also emerging that RNA sequence composition and structure are important determinants of the physical properties and function of biomolecular condensates[68]. In the context of our designed condensates, we observed that cation-π and π-π interactions between the polypeptide and RNA nucleobases enhance condensate viscoelastic behavior (Figs. 2 and 4). Similar modulation of condensate material properties may be achieved with RNA primary sequence variation leveraging the unique structural differences between purine (contain 10 π-electrons) and pyrimidine (contain 6 π-electrons) bases and/or their secondary structure[69]. Previous studies by our group reported that condensates formed by R/G-rich polypeptides and homopolymeric purine-containing RNAs [poly(A)] display substantially slower fusion speed as compared to homopolymeric pyrimidine-containing RNAs [poly(U)][44]. This was attributed to the presence of stronger short-range interactions (cation-π and π-π) between purine bases and R-rich IDRs[70]. Here we considered the possibility of altering the condensate dynamical properties by sequence patterning and secondary structure of RNA chains. To test this idea, we first used a single-stranded nucleic acid (ssNA, $G_5T_{30}C_5$) with a stem-loop structure, and compared the resultant condensate network dynamics with condensates formed by an unstructured nucleic acid dT40 (Fig. 6a, b). The dynamical properties of [RGPGG]₅-dT40 condensates are nearly identical to [RGPGG]₅-rU40 RNA with the viscous modulus $G''$ dominating the rheological response throughout the experimental frequency range (Fig. 6c). On the contrary, [RGPGG]₅-$G_5T_{30}C_5$ condensates displayed enhanced viscoelasticity with a network relaxation time of ~24 ± 9 ms (Fig. 6d). The viscosity of [RGPGG]₅-$G_5T_{30}C_5$ condensates is 1.9 ± 0.4 Pa.s, which is one order of magnitude higher than the viscosity of [RGPGG]₅-dT40 condensate (0.14 ± 0.03 Pa.s; Fig. 6e, f). These results showcase the effect of RNA sequence composition and secondary structure on the viscoelasticity of biomolecular condensates. We next tested another structured ssNA consisting of five GGGGCC repeats, (G4C2)₅, which is known to form stable G-quadruplex structures[71]. Increased numbers of the G4C2 repeats in the *C9orf72* gene have been genetically associated with neurodegeneration in amyotrophic lateral sclerosis (ALS) and frontotemporal dementia (FTD) patients[72]. At the same time, G-quadruplex structure forming NAs have previously been reported to interact with R/G-rich IDRs in ribonucleoproteins[73–75]. We observed that (G4C2)₅ NA formed spherical condensates in the presence of [RGPGG]₅ (Fig. 6g) albeit the condensates did not facilitate optical trapping of embedded beads likely due to their higher refractive index (hence lower refractive index mismatch with the beads) and density. To assess their dynamical behavior, we performed video particle tracking microrheology with 200 nm polystyrene beads (Fig. 6h). The MSD curves showed almost complete arrest of

embedded particles in timescales on the order of 5 s (Fig. 6h), followed by a slow sub-diffusive motion ($\alpha$ ~ 0.86). This observation indicate that [RGPGG]₅-(G4C2)₅ condensates contain an elastic network which flows very slowly ($\tau_M$ > 5 sec). This behavior is opposite to [RGPGG]₅-dT40 condensates, which showed a linear MSD curve ($\alpha$ ~ 0.99) without any detectable dynamical arrest of probe particles (Figs. 6h and 2d). Based on these results, we conclude that the LVE properties of polypeptide-RNA condensates are strongly influenced by the RNA sequence composition, patterning, and secondary structure.

## Discussion

Quantitative understanding of the molecular determinants of biomolecular condensates' viscoelastic properties is important for two reasons: (i) the knowledge of the condensate physical behavior at different timescales (i.e., viscous vs. elastic regime) can provide invaluable insights into their functional regulation and their pathological liquid-to-solid transformation, and (ii) the establishment of a sequence-phase behavior-material property paradigm for disordered biopolymers akin to sequence-folding paradigm for globular proteins can enable building predictive models of biomolecular condensates and rational designing of synthetic membraneless organelles with programmable phase behavior and material properties. A growing number of recent studies indicate that biological condensates are viscoelastic fluids, and therefore exhibit distinct mechanical behavior at different timescales based on their network reconfiguration time (i.e., the making and breaking of physical crosslinks)[22,30,31,33]. Although, several theoretical works have attempted to quantitatively link the viscoelastic behavior of the condensates and the molecular structure of their constituents[22,76–78], experimental studies on the same remain sparse due to limited suitable methods that can access the material properties within the minute volume of the condensed liquid phase. Recently, active oscillatory microrheology using a dual-trap optical tweezer[30,31] was used to explore protein condensate maturation over time. It was found that homotypic protein condensates' aging manifests in slowing down the dynamical motion within the condensates rather than increasing the elasticity of the condensates. Another recent work utilized similar methods to confirm the viscoelastic nature of biomolecular condensates and to point to the role of shear stress relaxation in controlling the condensate dynamics and fusion behavior[79]. The present study, however, focuses on linking the viscoelastic properties of polypeptide-RNA condensates to the sequence and structure of their constituent biopolymers by implementing passive microrheology with optical tweezers (pMOT). This is achieved through systematically exploring the frequency-dependent LVE properties of generic IDP-RNA condensates formed by RNP-inspired sticker-spacer polypeptides. Our results reveal that R/G-repeat polypeptide-RNA condensates resemble Maxwell fluids, which behave as an elastic solid at short timescales and a viscous fluid at long timescales. Both viscosity and the time-dependent network dynamics can be precisely controlled via polypeptide chain sequence, RNA sequence composition and secondary structure. These findings add significantly to the growing interest in characterizing and engineering rheological properties of biomolecular condensates by presenting a set of sequence analytics.

We speculate that the sequence heuristics obtained from our designer polypeptides can provide insights into the condensate network properties formed by RGG-domains of natural RNPs, which display a great deal of sequence and length variations. For example, the RGG-domains of FET proteins have diverse sequence features such as the occurrence of PGG motifs in EWS and YGG motifs in TAF15 (Supplementary Fig. 13a). Based on

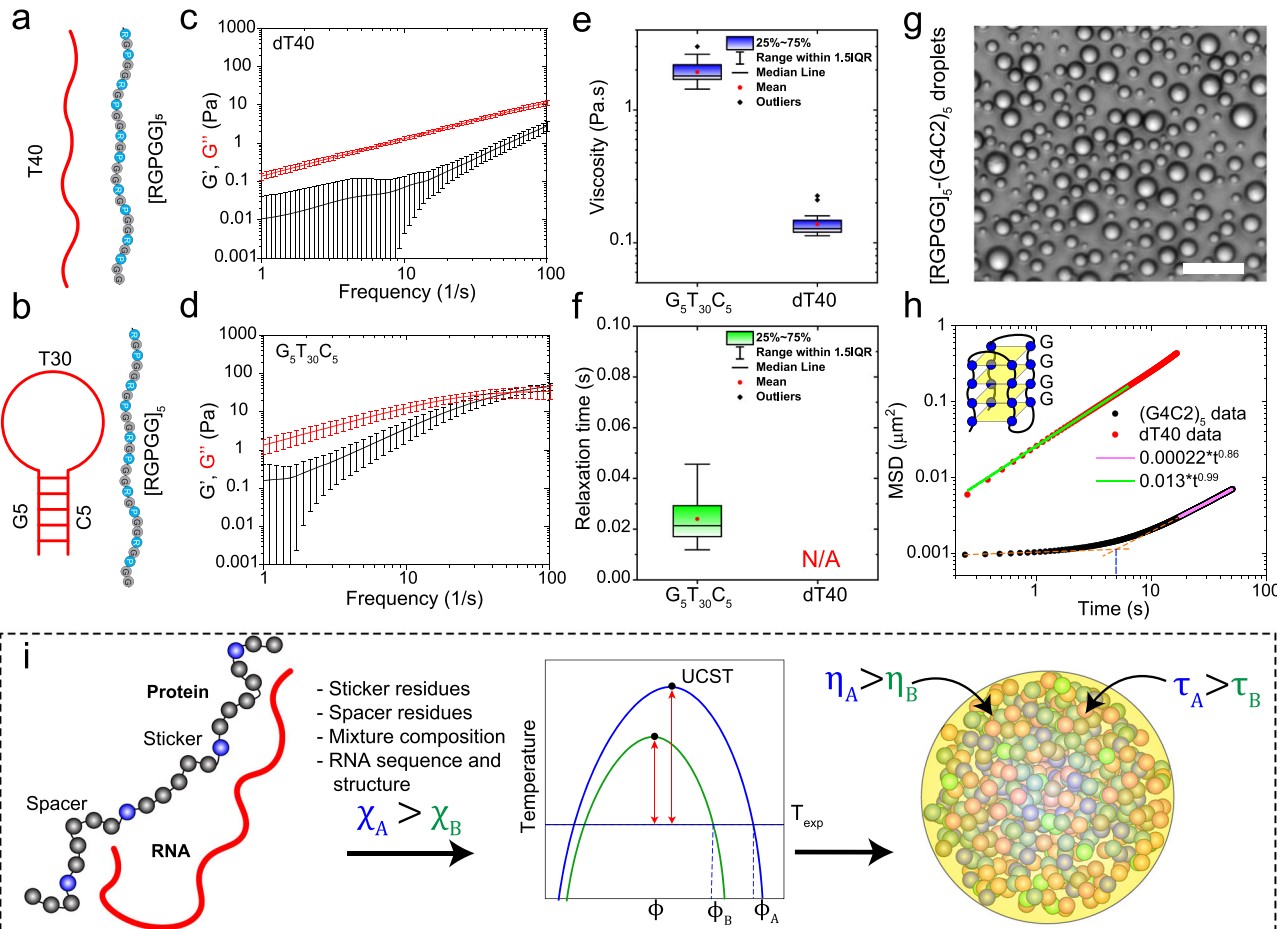

**Fig. 6 Effect of RNA sequence and structure on the LVE properties of peptide-RNA condensates. a** A scheme showing the sequence and expected disordered structure of the peptide and dT40. **b** A Scheme showing the sequence and expected structure of $G_5T_{30}C_5$. **c** Average viscoelastic moduli of condensates formed by [RGPGG]$_5$ and dT40 as measured by pMOT. Error bars represent one standard deviation (±1 s.d.). **d** Average viscoelastic moduli of condensates formed by [RGPGG]$_5$ and $G_5T_{30}C_5$ as measured by pMOT. Error bars represent one standard deviation (±1 s.d.). **e** Zero-shear viscosity of [RGPGG]$_5$-ssNA condensates for both structured ($G_5T_{30}C_5$) and unstructured (dT40) NA as calculated from the data in **b** and **d**. **f** Terminal relaxation time of [RGPGG]$_5$-NA condensates for both structured ($G_5T_{30}C_5$) and unstructured (dT40) NA as calculated from the data in **b** and **d**. For the data in **c–f**, the sample size is n=(22 and 20) measurements over 3 independent samples for (dT40 and $G_5T_{30}C_5$), respectively. **g** Brightfield image of condensates prepared by mixing [RGPGG]$_5$ and $(G_4C_2)_5$ NA at 5.0 mg/ml and 2.5 mg/ml concentrations, respectively. Scale bar is 20 μm. **h** Mean squared displacement (MSD) of 200 nm polystyrene particles within condensates formed by [RGPGG]$_5$ peptide with dT40 (red) and $(G_4C_2)_5$ (black). This data show that $(G_4C_2)_5$ forms condensates that have substantially slower network dynamics as compared to the condensates formed by dT40. The orange dashed lines extrapolate the two behaviors (flat MSD at short lag-times and increasing MSD at long lag-times) and the blue line indicates the extrapolated transition point at ~5 s. **i** A scheme summarizing the relation between the strength of intermolecular interactions and LVE properties of peptide-RNA condensates. Comparison between two idealized systems A (blue) and B (green) is shown with interaction parameter $\chi_A > \chi_B$. Due to stronger interactions in system A, the difference between the phase separation temperature (blue curve) and the experimental temperature is greater, leading to a deeper quench within the two-phase regime (red double-sided arrows). Additionally, stronger interactions in system A cause higher viscosity $\eta$ and a slower bond reconfiguration time $\tau$, which retards the network flow.

our results with synthetic condensates (Fig. 4a), we expected that EWS-RGG will have the fastest dynamics and least viscoelasticity due to the presence of multiple PGG motifs. Additionally, since TAF15-RGG has several YGG repeats as compared to FUS-RGG, we hypothesized that TAF15-RGG condensates will display a stronger viscoelastic response than FUS-RGG. To test this idea, we measured the dynamical properties of condensates formed with representative RGG polypeptides from FUS, EWS, and TAF15 with a single-stranded nucleic acid (dT40), and observed that indeed EWS$^{RGG}$ condensates displayed the weakest viscoelastic behavior ($\eta$~2 Pa.s, $\tau_M$ ~ 20 ms) whereas TAF15$^{RGG}$ condensates showed the strongest viscoelastic behavior ($\eta$ ~ 20 Pa.s, $\tau_M$ ~ 300 ms; Supplementary Fig. 13).

In addition to polypeptide sequence variations in a two-component system, mixing multiple polypeptide chains with distinct sticker-spacer architectures, and/or RNA base composition and structure can also tune the LVE properties of condensates (Figs. 5 and 6), giving access to multiple routes of regulation. A rationalization of the observed variations in condensate dynamical properties comes from the consideration that inter-chain interactions dictate the thermodynamic and rheological properties of network fluids[29]. Using phase diagram analysis, diffusion measurements, and MD simulations, we found that the LVE properties of the dense phase are correlated with the relative strength of intermolecular interactions between the polypeptide and RNA chains. In our phase diagram analysis (Fig. 4c), we observed a strong correlation between the stability of

condensates against salt and temperature and the linear viscoelasticity of the condensate network. Such a correlation can be understood using Flory-Huggins model of a phase separating mixture with UCST[61,62] (Fig. 6i). The mean-field intermolecular interactions between peptide-RNA complexes can be represented using the Flory parameter, $\chi$. For simplicity, we consider two systems, A and B, with corresponding interaction parameters $\chi_A$ and $\chi_B$ ($\chi_A > \chi_B$; Fig. 6i). Flory-Huggins theory predicts that UCST of system A will be higher than system B if $\chi_A > \chi_B$ and vice versa (Figs. 6i and 4h). Therefore, at a common experimental temperature that is lower than the phase separation temperature of both systems, system A will be positioned at a larger quench depth (defined as the difference between the phase separation temperature and the experimental temperature) than system B (Fig. 6i). Without ruling out any other effects, the polymer density in the condensed phase of system A will be larger than the polymer density of system B in the dense phase. Polymer dynamics in the dense phase, including network relaxation time scales and the rheological properties, are often linked to polymer density and inter-chain bond relaxation times[77]. For example, complex coacervates formed by oppositely charged polymers were previously reported to exhibit slow Rouse-like relaxation modes that are described by a sticky Rouse polymer dynamics model[80]. In such a case, the viscosity of the condensate is proportional to the polymer density within the condensates. Moreover, the network relaxation dynamics are governed by the timescale of bond formation and breakage. We note that describing a multi-component peptide-RNA system solely in terms of a global Flory parameter ($\chi$) is an oversimplification. Nevertheless, the particular choice of the model does not alter the drawn conclusions since most polymer UCST phase separation theories are generally consistent with the idea that stronger inter-chain attractions lead to denser condensed phases and that higher polymeric density leads to slower dynamics of the network structure due to steric forces and longer-lived physical bonds[61,81,82]. This explanation is consistent with our MD simulation results (Fig. 4d–h). Therefore, changing intermolecular interactions can regulate the rheological properties of the condensate through altering the polymer density within the condensed phase as well as the average bond lifetime (Fig. 6i).

In summary, our work reports that the dynamical properties of a phase separated condensate depend on the nature and strength of inter-chain interactions. The material properties of biomolecular condensates are thought to be critical to their physiological function and arrested phase separation (or liquid-to-solid transitions) have been linked to pathological dysfunction[19,83]. Laser tweezer-based microrheology of reconstituted RNP condensates can provide a deeper understanding of the context-dependent regulation of condensate material properties and enable the examination of various endogenous factors (such as post-translational modifications and genetic mutations) and exogenous effectors (small molecules) in manipulating their physical properties. Finally, engineered condensates formed by sticker-spacer polypeptides with programmable viscoelasticity can inspire new routes to design IDP-based soft biomaterials with tunable mechanics and provide a suitable platform to investigate how biochemical reaction dynamics are regulated by the condensate viscoelasticity.

## Methods

### Peptide and RNA samples
See Table-S1 for a list of peptides used in this study. All peptides ([RGRGG]$_5$, [KGKGG]$_5$, [RGYGG]$_5$, [RGPGG]$_5$, [RGFGG]$_5$, [RGSGG]$_5$, [RARAA]$_5$, [RQRQQ]$_5$, [RPRPP]$_5$, [RLRLL]$_5$, RG1Y, RG3Y, FUS-RGG, TAF15-RGG, and EWS-RGG were synthesized by GenScript USA Inc. (NJ, USA, >90% purity). All peptides contained a cysteine at the C-terminal for site-specific fluorescence labeling. Peptides were reconstituted in RNase-free water (Santa Cruz Biotechnology) containing 50 mM dithiothreitol (DTT) to prevent

cysteine oxidation and the stock solutions were split into multiple aliquots and stored at −20 ºC. Homopolymeric RNA [rU]$_{40}$ (MW 12,185 Da), ssDNA [dT]$_{40}$ (MW 12,106 Da), G$_5$T$_{30}$C$_5$ and (G4C2)$_5$ ssNA were purchased from Integrated DNA Technologies (IDT) and reconstituted in RNase-free water. All peptide and RNA stock solutions were checked under the microscope to ensure complete solubilization and the absence of aggregates. No additional purification was performed on the peptides or the RNA.

### Peptide-RNA condensate preparation
Each type of condensate was prepared by mixing peptide and RNA in a buffer containing 25 mM Tris-HCl (pH 7.5), 25 mM NaCl and 20 mM DTT. The peptide final concentration was 5.0 mg/ml and the rU40 RNA concentration was 2.5 mg/ml, corresponding to a mass ratio of 0.5. This ratio was chosen based on solution turbidity measurements of [RGRGG]$_5$-rU40 mixtures which showed maximum absorbance[41] (indicating a maximum degree of condensation; Supplementary Fig. 14a). All peptides underwent LLPS with RNA at this ratio and resulted in the formation of condensates which were then subjected to microrheology experiments to measure their linear viscoelastic behavior. Due to the comparative nature of this study, all buffer conditions were kept identical. For microrheology experiments, yellow-green carboxylate-modified polystyrene beads (1 μm in diameter, FluoSpheres™, Invitrogen) were added to the buffer at a low concentration (0.0003 % solids). These beads were found to partition well within peptide-RNA condensates[41]. For experiments performed with G$_5$T$_{30}$C$_5$ and (G4C2)$_5$ ssNAs, the same buffer was used except for the addition of 2.5 mM MgCl$_2$. For FET-RGG condensates (TAF15, EWS, and FUS), the polypeptide concentration was 10 mg/ml, the ssDNA T40 concentration was 1 mg/ml, and the buffer contained 25 mM Tris-HCl (pH 7.5), 25 mM NaCl and 20 mM DTT.

### Passive microrheology with optical tweezers (pMOT)
The method of passive microrheology used in our study was introduced by Tassieri, Preece, and Evans[37–40] and is built on the work on optical tweezer-based microrheology by Mason and Weitz[35]. The complex modulus of a material (condensate) is calculated from the analysis of the position fluctuations of an embedded probe particle trapped with an optical tweezer. The motion of a trapped bead within the medium is mediated by the thermal fluctuations of the medium and constrained by the harmonic potential of the optical trap. This is one of the few microrheology methods that do not require an independent optical trap calibration procedure[37,84–87].

Condensate-forming samples (5 μl) of peptides and RNA were placed on a Tween20-coated microscope coverslip (0.17 mm thickness, 18×18 mm dimensions). Next, a glass slide with two double-sided tape strips was gently placed on the sample, sandwiching it between the coverslip and the glass slide. ~100 μl of mineral oil was injected into the chamber between the coverslip and the glass slide such that oil surrounds the sample from all directions. This was done to ensure that the sample does not experience evaporation or evaporation-induced flow. The condensate sample was left to equilibrate for a minimum of 1 h. The pMOT experiments were not initiated until all droplets were settled on the coverslip surface and no droplet fusion events were observed. To measure the frequency-dependent viscoelastic moduli, the custom-made flow chamber was loaded onto a correlative confocal microscope-optical tweezer setup (LUMICKS C-Trap) with a 60x water immersion objective and a bright-field camera. Next, the optical trap was used to trap a bead that is embedded within a peptide-RNA condensate. The trapping power was set to ~100 μW initially to trap the bead. Using the optical trap, the bead was then positioned at the center of the condensate and ~2–5 μm above the coverslip surface. The trapping power was lowered to the minimum value that ensures constraining the bead within the optical trap. The trap-constrained motion of the bead was tracked using the bright-field camera with a 500 Hz acquisition rate. The trajectory of the bead (X-Y coordinates as a function of time) was extracted using the built-in tracking algorithm in the instrument (LUMICKS C-trap). For each peptide-RNA system, we collected trajectories from 3–5 condensates for three independently prepared samples (a total of 9–15 trajectories). Every single trajectory was obtained by tracking the bead for a minimum of 10 min and a maximum of 45 min depending on the peptide-RNA system and the quality of the extracted autocorrelation curves (see the data analysis section). All trajectories were analyzed using custom-built python scripts to obtain frequency-dependent viscoelastic moduli and viscosity. Trajectories that showed asymmetric motion in the X-Y direction were discarded. The analysis of the pMOT experiments as well as additional control measurements are discussed in detail in Supplementary Note 1.

### Video particle tracking microrheology (VPT)
200 nm yellow-green carboxylate modified polystyrene beads (FluoSpheres, Invitrogen) were used for VPT measurements. The samples were prepared at identical conditions as described in the sample preparation section. Fluorescence video imaging was done using a Zeiss Primovert inverted microscope equipped with a 100x oil immersion objective and a Zeiss axiocam 503 monochrome camera. Movies of the beads diffusing within the condensate were collected for approximately 2–5 min. Particle tracking was performed using Trackmate[88] plugin in Fiji-ImageJ. Mean squared displacement (MSD) was calculated from the trajectories for several lag times $\tau$ using home-built python scripts. The ensemble average MSD was calculated (from ~20–100

individual particles). In general, the MSD scales as a power law with time ($MSD \propto t^{\alpha}$), where $\alpha$ is diffusivity exponent. In the special case when $\alpha = 1$ (as in [KGKGG]$_5$-rU40 condensates), the MSD can be fitted to extract the diffusion coefficient ($MSD = 4D\tau + b$). In this equation, D is the diffusion coefficient and b is a constant accounting for the noise. The condensate viscosity $\eta$ was then calculated from the Stokes-Einstein equation[41]

$$\eta = \frac{k_B T}{6\pi D a} \quad (2)$$

Where $k_B$ is Boltzmann constant, $T$ is the temperature, and $a$ is the particle radius. For [RGYGG]$_5$-rU40 and [RGRGG]$_5$-rU40 condensates, the MSD was fitted with $MSD = d^2(1 + \frac{\tau}{\tau_c})$, which is the predicted behavior of a Maxwell fluid[31]. For each condensate type, we collected trajectories from 3–5 condensates in 3 independently prepared samples. Each condensate contained somewhere between 20-100 microspheres. Trajectories of beads close to the condensate surface were excluded from the MSD calculation.

**Fluorescence recovery after photobleaching (FRAP).** Peptide-RNA condensates were prepared using 5.0 mg/ml peptide and 2.5 mg/ml rU40 RNA mixed in 25 mM Tris-HCl (pH 7.5), 25 mM NaCl, 20 mM DTT containing ~500 nM of FAM-labeled RNA oligo rU10 (purchased from IDT). Samples were injected in a Tween20-coated imaging chamber and loaded to the confocal microscope stage (LUMICKS C-trap). Bleaching was achieved using a 488 nm laser at 100% laser power for ~0.5 seconds. The fluorescence intensity of a given region of interest (ROI) was then recorded until the recovery was complete. The bleaching ROI size and shape were kept identical across all samples. The intensity time traces were plotted for comparison and extraction of $t_{1/2}$ values. For each peptide-RNA sample, 5 recovery traces were collected and averaged. Intensity error bars were estimated using the standard deviation at each recorded time point.

**Temperature-salt state diagram measurements.** Peptide-RNA condensates were prepared using 5.0 mg/ml peptide, 2.5 mg/ml rU40 RNA mixed in a buffer (25 mM Tris-HCl, pH 7.5 and 20 mM DTT) with the desired salt concentration as noted in Fig. 4 in the main text. Peptide-RNA samples were then sandwiched between a PEG5000-coated glass slide and a coverslip. The sample was surrounded by mineral oil to prevent sample evaporation due to heating. The sample was then placed into a custom-built temperature stage (Instec Inc., temperature range: 5–90 °C) and loaded on a Zeiss Primovert inverted microscope equipped with a 40x objective and a Zeiss axiocam 503 monochrome camera. To determine the phase separation temperature ($T_{ph}$), the sample was heated first to maximum temperature (90 °C) until all condensates were dissolved. Next, the sample was cooled down in steps of 5 °C and an equilibration time of 5 min until LLPS was observed. Once the LLPS transition was noted, samples were heated again until condensates were dissolved and the LLPS temperature was subsequently approached in steps of 2 °C and equilibration time of 5 min in each temperature to determine the $T_{ph}$.

**Mesh size determination experiments.** Peptide-RNA samples were prepared at 5.0 mg/ml peptide and 2.5 mg/ml rU40 RNA in a buffer containing 25 mM Tris-HCl (pH 7.5), 25 mM NaCl and 20 mM DTT. After mixing the peptide and the RNA and the formation of condensates, a small concentration (~500 nM) of tetramethylrhodamine-labeled dextran of desired size was added to the sample. The sample was mixed and subsequently imaged under a confocal microscope (LUMICKS C-trap). If the hydrodynamic radius of the dextran molecule is smaller than the mesh size of the condensate, it is expected that dextran molecules will be positively recruited into the condensates. However, if the hydrodynamic radius of the dextran molecule is larger than the mesh size of the condensates, dextran recruitment will not occur. The partition behavior of dextrans of variable molecular weights (and variable hydrodynamic radii) was recorded and the mesh size range was determined accordingly[42,57]. The smallest dextran molecule that partitioned negatively in the condensate signifies the upper limit of the mesh size.

**Glass slide and coverslip preparation.** For the temperature-salt state diagram, mPEG5K-Silane was used to coat glass slides and coverslips to prevent condensates from sticking to the glass surfaces. The PEG coating procedure was performed according to the description of Alberti and coworkers[89]. Briefly, glass slides were incubated in 2% Hellmanex solution for two hours and then rinsed with milliQ water and dried under compressed airflow. Next, the glass slides were immersed in a solution containing 20 mg/ml of methyl-PEG5K-Silane and 100 mM HCl dissolved in toluene and incubated for 18 h in a closed glass staining jar. Glass slides were then rinsed with toluene, 90% ethanol, and MilliQ and dried under compressed airflow.

For all other experiments in this work, Tween20 coating was used to prevent droplet spreading on the glass surface. Glass slides and coverslips were first cleaned with 70% ethanol and dried under compressed airflow. Next, glass slides and coverslips were immersed in a 20% vol/vol solution of Tween20 for 30 min. Subsequently, glass slides and coverslips were rinsed 6–7 times with MilliQ water and dried using compressed air. Finally, glass slides and coverslips were dried in an oven set at 40 °C for 4–8 h and stored at room temperature for later use.

**Bead Halo assay.** Samples were prepared by mixing [RGRGG]$_5$ with rU40 RNA at 5.0 mg/ml peptide and 2.5 mg/ml RNA concentrations in a buffer containing 25 mM Tris-HCl (pH 7.5), 25 mM NaCl, and 20 mM DTT. Carboxylate-coated beads were added to the buffer along with the bait molecule (either [RGRGG]$_5$-A594 or rU10-FAM) before mixing the peptide and RNA. The bait molecule concentration was kept at ~200 nM. Upon mixing the peptide and the RNA, condensates were formed. The sample was thoroughly mixed and then sandwiched between a coverslip and a microscope glass slide spaced by two layers of double-sided tape. The sample was kept to equilibrate for 10 min and then loaded onto a confocal microscope (LUMICKS, C-trap). Several bead-harboring condensates were imaged using both brightfield and fluorescence illumination. For each bait molecule ([RGRGG]$_5$ or rU10), two independent samples were prepared and several condensates were imaged. Intensity profiles were plotted to inspect the fluorescence intensity at the bead surface and compare it with the intensity inside the droplet.

**All-atom simulations.** The explicit all-atom (AA) simulations with model tri-nucleotide and tri-peptide systems were done with OpenMM 7.5 using Amber a99SB-disp*-ildn force field for biomolecules and TIP3P model for water. The system consisted of tri-peptides and tri-nucleotides packed in a cubic box with a concentration of ~100 mg/ml, solvated, and charge neutralized with Na+ and Cl− ions at 10 mM concentration at peptide:RNA ratio of 1:2. The total number of atoms was ~100,000. In total 10 independent replica simulations were run for 1 microsecond each. The combined trajectory from the 10 replicas was used for deriving the potentials of mean force by inverting the radial distribution functions of monomeric units. For each simulation trajectory, integration time was set to 2 fs. After minimization, the simulations were done in an NPT ensemble where the pressure was kept at 1 atm via Monte Carlo Barostat and the temperature was kept at 300 K using a standard Langevin integrator with 1/ps set for the friction coefficient.

**Phase coexistence simulations.** The simulations with coarse-grained (CG) models of protein and nucleic acid chain mixtures were done using HOOMD-Blue 2.9.6 molecular dynamics library. We used a 1 bead per residue/base model for amino acids and RNAs based on the formulation first presented in Alshareedah and coworkers[42]. The short-range potentials were calibrated according to the all-atom simulations to account for the differential stickiness of amino acids. Slabs were made by first randomly packing polypeptide and RNA chains in a cubic box. The box was then compressed to ~25 nm over the course of $10^5$ steps at a high temperature corresponding to T = 600 K. The z-axis of the box was then extended to ~125 nm, and the packing was equilibrated for $10^6$ steps to allow the dense and dilute phases to form while quenching the system to the target temperature. Finally, the equilibrated dense and dilute phases were simulated for an additional $10^7$ steps to collect data. Simulations were run at various temperatures with 4000 polypeptide chains and an appropriate number of RNA chains ensuring a net chain charge neutrality. In all simulation steps, a Langevin integrator was used with a friction coefficient of 0.01.

**Bioinformatics analysis.** Amino acid sequences of RNA-binding proteins were downloaded from Uniprot (RefSeq) based on the annotation provided in RBPDB[54]. RG/RGG motifs within these proteins were extracted using a custom python script. RG/RGG motif was defined as per Thandapani et al.[52]. Briefly, peptide motifs with the sequences RG-$x_{(0-4)}$-RG and RGG-$x_{(0-4)}$-RGG were extracted, where $x$ can be any amino acid. A binary presence/absence test was performed to determine how many of the extracted sequences contain each of the 20 amino acids within the spacer [i.e., $x_{(0-4)}$]. Results are presented as a bubble chart in the main text Fig. 2c, where the diameter of an individual bubble corresponds to the fraction of spacers that contain respective amino acids. Cyan bubbles represent fractions > 10%.

**Software.** pMOT analysis was done using custom-made python scripts. Fiji-ImageJ[90] (version 1.52p) was used for image processing. OriginPro (2018b) was used for Graphing. Adobe Illustrator CC (2019, v23.0) was used for the figure assembly and production. ZEN (blue, v2.3) was used for image recording using a Zeiss Primovert microscope. Bluelake (v1.6.11) was used for image recording and particle tracking using Lumicks C-Trap microscope. HOOMD-Blue 2.9.6 was used for coarse-grained molecular dynamics simulations and OpenMM 7.5 was used for all-atom simulations. Ovito 3.5 was used for generating phase coexistence snapshots and nglview 3.0 was used for visualizing all-atom residue-base interaction snapshots shown in Fig. 4.

**Statistics and reproducibility.** All experiments reported in this manuscript have been repeated at least three times with consistent results. The viscosity measurements are repeated for at least 12 data points from different condensates across three independently prepared samples. The relaxation times are obtained from at least 12 data points from different condensates across three independent sample preparations. The VPT experiments were replicated three times across different sample preparations. Bright-field and fluorescence microscopy images are representative of a large set of images exhibiting similar features. For temperature-salt state diagrams, random points were selected and reproduced three times successfully. FRAP experiments were repeated five times (within 1-2 sample preparations)

for each peptide-RNA system. Microscopy images in Figs. 1a, 4b, 6g; and Supplementary Figs. 2, 6, 7, 8, 11a, b, 18b, 19 have been replicated at least 3 times for independent samples and represent a larger set of images with identical features.

**Reporting summary**. Further information on research design is available in the Nature Research Reporting Summary linked to this article.

## Data availability

All data relevant to the findings of this manuscript are included in the manuscript, the supplementary appendix and the source data file. Additional raw data corresponding to the viscoelasticity measurements using pMOT are deposited in GitHub (https://github.com/I-Alshareedah/pyMOT/tree/main/Data). Source data are provided with this paper.

## Code availability

All analysis and calculations were done using custom-made python scripts that are available on GitHub (https://github.com/I-Alshareedah/pyMOT/tree/main/Code).

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

## Acknowledgements

The authors acknowledge Dr. Manlio Tassieri of the University of Glasgow, UK for valuable discussions on the application of passive microrheology with optical tweezers. P.R.B. acknowledges the College of Arts and Sciences at the University at Buffalo, SUNY, and the National Institute of General Medical Sciences (NIGMS) of the National Institutes of Health (R35 GM138186) for financial support. D.A.P acknowledges financial support from College of Liberal Arts and Sciences at the Iowa State University and the National Institute of General Medical Sciences (NIGMS) of the National Institutes of Health (R35 GM138243).

## Author contributions

P.R.B. conceived the idea. P.R.B. and I.A. designed the study. I.A. performed the experiments and data analysis. M.M.M. performed the bioinformatics analysis. M.P. and D.A.P. performed all-atom and coarse grained MD simulations. P.R.B. and I.A. wrote the manuscript with input from M.M.M. and D.A.P.

## Competing interests

The authors declare no competing interests.
