## [Peer Review File · Nature Communications]

Programmable Viscoelasticity in Protein-RNA Condensates with Disordered Sticker-Spacer PolypeptidesReviewers' Comments:

Reviewer #1:

Remarks to the Author:

This paper presents a careful, systematic study on the viscous and elastic moduli of condensates formed by peptide and RNA mixtures. The authors demonstrated that they could empirically tune the zero-shear viscosity and the crossover frequency. I have the following major concerns:

1. While the experimental technique and the resulting viscoelastic data are interesting, I'm hard pressed to find how the data can be used for anything useful, e.g., explaining something about peptide-RNA condensates that's now possible because of this study. It is not very clear how this paper would influence thinking this field. The authors seem to concede as much, with general statement like:

"The LVE properties of biomolecular condensates are thought to be intricately coupled to their physiological function and alterations thereof have been linked to pathological dysfunction^{4,32}. Therefore, the observed tunability of the LVE behavior of peptide-RNA condensates may provide key insights into the context-dependent regulation of material properties of intracellular RNP condensates"

What would the key insights be? I hope this comment will motivate the authors to seriously think about explicitly connecting their work with other studies to develop a deeper picture about biomolecular condensates.

2. Dividing viscoelastic materials into a viscous regime and an elastic regime, based on whether G'' is greater than G' , is too simplistic. This classification is valid only in the limits where G'' is much greater than G' or G'' is much less than G' . For much of the frequency range, both G'' and G' are important and hence the condensates are viscoelastic, i.e., neither viscous nor elastic -- this is especially true for condensates such as [RGYGG]₅ that follows the Jeffrey model. The authors need to rethink how to present the data in the text and in the figures, as their current presentation clearly misleads authors.

3. The authors made some attempts to provide some molecular explanation for their data, but the explanation leaves much to be desired. E.g., in commenting on the much longer τ_M value of RGYGG₅ relative to the counterparts of RGRGG and RGFGG, they stated "This is consistent with the fact that Tyrosine residues are stronger drivers of phase separation as compared to Phenylalanine^{13,14}" One would think R is an even stronger driver for their peptide-RNA system. Clearly they need to come up with a consistent rationale.

Other issues:

4. Their "frequency" is likely angular frequency (ω), to be distinguished from frequency (f). The two are related by a factor 2π . ω is in units of radian per second, whereas f is in units of Hz.

5. The data in Fig. 2i needs to be analyzed in some theoretical model to show that the mid-point of the transition is indeed mostly determined by τ_M .

6. Eq. 12 in Supplementary Information is missing an "a" in the denominator. More seriously, the MSD data can be used to deduce the viscosity via the Stokes-Einstein relation only when the exponent α is 1. When α differs from 1, D does not even have the correct unit (i.e., $\mu\text{m}^2/\text{s}$). This may mean that the authors need to reassess all their VPT data for viscosity.

Reviewer #2:

Remarks to the Author:

Banerjee and co-workers use optical tweezers microrheology to characterize the viscoelastic properties of programmable protein-RNA condensates. The overarching claim is that they can tune the viscoelastic properties of the condensates over several orders of magnitude by varying the molecular architecture of the sticker-spacer polypeptides used in the aforementioned condensates. I find the concept intriguing and relevant to biological and engineering applications. However, there are essential controls, experiments, and details in regards to the measurement of viscoelastic properties that are needed to be able to properly evaluate the results and ensure the results are not artifacts of the experiment design. My main concerns are detailed below:

1. The distance from the glass slide used in pOTM measurements is $\sim 2-5$ μm . This is not enough distance from the surface to be able to ignore surface effects on the trapping signal. One needs to be at minimum ~ 10 bead diameters from a surface to neglect this effect. The authors show that being in the center of a condensate versus the edge does not significantly impact the results but the soft condensate interface is quite different than that of the glass slide. The authors need to demonstrate that at distances larger than 2-5 μm from the surface their results are the same. Or they need to account for surface effects and subtract them from their signal.

2. The authors state they use carboxylated beads but there is no mention of surface passivation. Carboxylated beads that are not coated with another protein or polymer are sticky and likely interact with the protein-RNA network. This interaction will have a major impact on the results presented. For the viscoelastic properties to be accurately measured the beads can only have steric interactions with the network with no sticking. Authors need to demonstrate that the beads are not adhering to the network or need to repeat measurements with beads coated with BSA, PEG or another inert polymer (as is standard in OTM microrheology).

3. Another issue that is not addressed is the small bead size (1 μm diameter?). How does the bead size compare to the mesh or pore size of the condensate network? It seems that the varying viscoelastic properties measured could easily be a result of changing network architecture that changes the pore size. If the bead is not sufficiently larger than the pore size of all the condensates tested then the measured effects can not be considered to represent the viscoelasticity of the condensate. Authors need to show evidence of or calculations of the pore size and show that the bead size is substantially larger than that. Alternatively, they could perform measurements using varying bead sizes (larger) and show that the results are not affected by changing bead sizes.

4. All of the viscoelastic data presented is very (suspiciously) clean. Yet there is no mention of smoothing. Is this really the raw data presented without smoothing? It seems that with only 9-15 trials and with biological samples there would be more noise in the signal. Especially, in Fig S9 where individual trials are plotted. Please comment.

5. Authors should show fits of the autocorrelation functions with multi-exponential functions (described in SI) to assess goodness of fit. I suspect that the very clean data I discussed in #4 is a result of this fitting but its not clear. It's not clear what the actual data would look like without relying on the fit. This delineation needs to be made clear

6. In pOTM experiments authors use video particle tracking to measure thermal fluctuations. This method is inherently lower accuracy than using the laser signal. Why was this chosen? And what is the centroid positioning accuracy? This data needs to be shown and uncertainty provided.

7. Related to #6 why is the laser signal used for the trap flipping experiments and video tracking used for pOTM measurements? What is the DAQ rate for the laser signal?

8. I'm struggling to see the point of the 'step-strain' flipping experiment. The presented timescales do not appear to correlate with any physically relevant relaxation mechanism or timescale and do not seem to provide any additional information. I also do not see how this measurement is an

'independent verification' of results? This measurement, like pOTM, also depends on the size of bead relative to the condensate pore size. It also depends on the distance the stage is moved (which is not justified in the paper). If the authors want to use this type of analysis it needs to be unpacked more and these issues need to be addressed.

9. The closeness of their measured water viscosity to reported values is not great (>20% difference). This makes me question their methods. One reason for this discrepancy could be surface effects (being too close to the glass slide). Authors should check this.

Reviewer #3:

Remarks to the Author:

This is an interesting and very important study that adds significantly to the field and will have a noticeable impact. The manuscript is well-written and concise. Reported data are of utmost importance and definitely will be of great interest to many researchers. The authors conclude this exceptional study stating "the application of laser tweezer-based microrheology will pave the path for understanding the molecular origins of viscoelasticity in membrane-less organelles". I could not agree more with this assertion.

Vladimir N. Uversky

Point-by-point response to Reviewers' comments

Reviewers' comments are colored in **black**, the authors' response is colored in **blue**. Hyperlinked references that are cited in our responses are underlined. The major changes made are highlighted in **yellow** in the revised manuscript and listed in the context of individual comments of the reviewers below. Figures or figure panels reporting new data are included in this response for clarity and ease of access, figure captions are colored in **dark red**.

Reviewer #1 (Remarks to the Author):

This paper presents a careful, systematic study on the viscous and elastic moduli of condensates formed by peptide and RNA mixtures. The authors demonstrated that they could empirically tune the zero-shear viscosity and the crossover frequency. I have the following major concerns:

1. While the experimental technique and the resulting viscoelastic data are interesting, I'm hard pressed to find how the data can be used for anything useful, e.g., explaining something about peptide-RNA condensates that's now possible because of this study. It is not very clear how this paper would influence thinking this field. The authors seem to concede as much, with general statement like:

"The LVE properties of biomolecular condensates are thought to be intricately coupled to their physiological function and alterations thereof have been linked to pathological dysfunction(4,32). Therefore, the observed tunability of the LVE behavior of peptide-RNA condensates may provide key insights into the context-dependent regulation of material properties of intracellular RNP condensates"

What would the key insights be? I hope this comment will motivate the authors to seriously think about explicitly connecting their work with other studies to develop a deeper picture about biomolecular condensates.

We thank the reviewer for raising this point. Indeed, motivated by this comment, we have added significant new experimental data and discussions in the revised manuscript that we discuss briefly here in the context of "*what are the key insights obtained from this study*" and "*how this study may influence the thinking in this field*"? A number of recent studies have indicated that biomolecular condensates are complex fluids with viscoelastic properties. Examples of these are:

1- Woodruff JB, Hyman AA, & Boke E (2018) Organization and function of non-dynamic biomolecular condensates. Trends in biochemical sciences 43(2):81-94.

2- Choi J-M, Holehouse AS, & Pappu RV (2020) Physical principles underlying the complex biology of intracellular phase transitions. Annual Review of Biophysics 49.

3- Jawerth LM, et al. (2018) Salt-dependent rheology and surface tension of protein condensates using optical traps. Physical review letters 121(25):258101.

4- Jawerth L, et al. (2020) Protein condensates as aging Maxwell fluids. Science 370(6522):1317-1323.

5- Espinosa JR, et al. (2020) Liquid network connectivity regulates the stability and composition of biomolecular condensates with many components. Proceedings of the National Academy of Sciences 117(24):13238-13247.

These studies have also raised concerns about utilizing simple microscopy-based techniques such as fluorescence recovery after photobleaching (FRAP) alone to quantify the material properties of condensates, leading to a growing interest in utilizing suitable rheological methods to probe condensates dynamical properties across different time-scales. Therefore, quantifying the condensate network structure and dynamics, and understanding the molecular determinants that govern them is important and remains a key aspect of emerging research in the field. Moreover, the fluid properties of biomolecular condensates are altered in many disease conditions, and transitions from a viscous (or a viscoelastic fluid) state to a solid-like state are often correlated with the onset of pathology. This particular aspect has recently been highlighted by Pappu, Taylor and co-workers (Mathieu C, Pappu RV, & Taylor JP (2020), Science 370(6512):56-60). These observations and perspectives have collectively motivated us in conducting a systematic study on **understanding the molecular determinants of condensate viscoelasticity utilizing tractable model systems**. The purpose is to build predictive thermodynamic models of biomolecular condensates aiding the establishment of a sequence-phase behavior-material property paradigm for disordered biopolymers akin to sequence-folding paradigm for globular proteins.

In our study, we adopted passive microrheology with optical tweezers (pMOT) to quantify the viscoelastic properties of synthetic condensates formed by disordered sticker-spacer polypeptides and RNA. We note that the application of pMOT to characterize condensate dynamical properties is a methodological innovation in the field and this is only the third study on quantifying frequency-dependent viscoelastic properties of biomolecular condensates [after Jawerth, L. et al. (2020), Science 370, 1317-1323 and Jawerth L. et al. (2018), Physical review letters 121(25):258101]. Our results reveal that R/G-repeat polypeptide-RNA condensates resemble Maxwell fluids, which behave as an elastic solid at short timescales and as a viscous fluid at long timescales. Both viscosity and the time-dependent network dynamics can be precisely controlled via polypeptide chain sequence, RNA sequence composition and secondary structure (*The nucleic acid sequence and structure-dependent variation of condensate viscoelasticity is new additions to the revised manuscript*). These findings add significantly to the growing interests in systematically characterizing and rationally engineering rheological properties of biomolecular condensates by presenting a set of sequence heuristics. **To our knowledge, these observations are the first of their kind where sequence-encoded inter-chain interactions at the microscopic level are quantitatively linked to the dynamical properties of the condensate at the mesoscale.**

To establish that it is indeed the inter-chain interactions between polypeptide chains and RNA that govern the material properties of condensate network, *we now included results in the revised manuscript from complementary biophysical assays, including temperature-dependence of peptide-RNA phase separation (Fig. 4a-c), diffusion measurements by FRAP (Fig. S5), condensate mesh-size estimation (Figs. S6&S7), all-atom simulations (Fig. 4d-f) and phase coexistence simulations (Fig. 4g-h)*. Based on these results, we conclude that the sequence heuristics obtained from our designer polypeptides can provide insights into the condensate network properties formed by RGG-domains of natural RNPs, which display a great deal of sequence and length variations. For example, the RGG-domains of FET proteins have diverse sequence features such as presence of PGG motifs in EWSR1 and YGG motifs in TAF15 (Fig. S13a). Based on our results with synthetic RG/RGG-RNA condensates, we expected that EWSR1-RGG will be most dynamic and due to the presence of multiple PGG motifs within the sequence. Additionally, since TAF15-RGG has more YGG repeats than FUS-RGG, we expect TAF15-RGG to have a dominant viscoelastic response as compared to FUS-RGG. Indeed, our experiments revealed that EWSR1-RGG display the most dynamic behavior ($\eta \sim 2$ Pa.s, $\tau_M \sim 20$ ms) whereas TAF15-RGG is the least dynamic ($\eta \sim 20$ Pa.s, $\tau_M \sim 300$ ms). *These results are included in the revised manuscript (Fig. S13).*

In the revised manuscript, we have also included new experimental data revealing that in addition to polypeptide sequence variations, *RNA sequence composition and secondary structure strongly influence the dynamical properties of these condensates (Fig. 6 in the revised manuscript)*. For example, replacing the disordered rU40 RNA with RNA containing either a stem-loop or G-quadruplex structure resulted in viscoelastic transitions, with G-quadruplex NA showing a significantly stronger effect on the condensate viscoelasticity. *These newly added results provide additional support to the emerging role of RNA in regulating the physical properties of biomolecular condensates (Roden C & Gladfelter AS (2021), Nature Reviews Molecular Cell Biology 22(3):183-195)*.

In light of these new experimental results, we have provided a molecular/thermodynamic interpretation of our results in the revised manuscript linking intermolecular interaction strength with the phase behavior of the mixture and the viscoelastic properties of the dense phase. In our phase diagram analysis, we observed a strong correlation between the stability of condensates against salt and temperature and the linear viscoelasticity of the condensate network. For example, changing stickers from P-to-R-to-F/Y increases attractive interactions with RNA chains (Figure 4c-h), leading to a progressively higher phase separation temperature and slower dynamical properties of the dense phase. This observed correlation is consistent with the prediction of a polymer model of a phase separating mixture with upper critical solution temperature (See Fig. 6g). Briefly, stronger inter-chain attractions are expected to result in denser condensed phases, which is exactly what we observe in our phase coexistence simulations (Fig. 4g-h). Higher polymeric density typically leads to slower dynamics of the network structure due to steric forces and longer-lived physical bonds. Therefore, our observed correlation between the intermolecular interaction strength and the viscoelastic behavior of peptide-RNA condensates lends support to the notion that stronger interactions lead to denser condensates and longer-lived physical bonds with slower network dynamics. These results may provide a thermodynamic basis for point mutations in RNPs, such as G156E mutation in FUS, that has been shown to accelerate a liquid-to-solid phase transition (Patel, A. et al. (2015) Cell, 162(5), 1066-1077).

Therefore, considering the reviewer's comment, we have made substantial changes in the revised manuscript, which are enlisted below. Our "Results" and "Discussion" sections now reflect the utility of our results (as discussed here briefly) and connect them with the current developments in the field. With explanations provided here and the revisions made in our manuscript/ inclusion of additional experimental results and simulation, we hope that we are able to successfully address the reviewer's concern.

List of revisions made:

- 1- Temperature and salt-dependent LLPS of peptide-RNA mixtures to connect the variance in material properties to phase behavior and intermolecular interactions. Figure 4b&c; Figure S9a.
- 2- All-atom and phase coexistence simulations to provide a molecular explanation to the variable viscoelastic properties of peptide-RNA condensates. Figure 4d-h; Figure S9b.
- 3- Rheology measurements reporting the role of RNA sequence and shape in tuning the viscoelasticity of peptide-RNA condensates. Figure 6a-h.
- 4- Findings on synthetic peptide systems were extended to naturally occurring RGG domains of the FET family of ribonucleoproteins. Figure S13.
5. To provide a molecular/ thermodynamic interpretation of our results, a new section was added to the manuscript "*The LVE properties of peptide-RNA condensates are correlated with the strength of attractive interactions between polypeptide and RNA chains*"
- 5- A discussion on the significance of our findings is added to the "Discussion" section, this discussion summarizes the main points that we describe in this response.

2. Dividing viscoelastic materials into a viscous regime and an elastic regime, based on whether G'' is greater than G' , is too simplistic. This classification is valid only in the limits where G'' is much greater than G' or G'' is much less than G' . For much of the frequency range, both G'' and G' are important and hence the condensates are viscoelastic, i.e., neither viscous nor elastic -- this is especially true for condensates such as [RGYGG] 5 that follows the Jeffrey model. The authors need to rethink how to present the data in the text and in the figures, as their current presentation clearly misleads authors.

It is true that these materials are not viscous or elastic but they are viscoelastic. However, when looking at the frequency range, there are clearly two regimes, a regime where the behavior is **dominantly elastic** ($G' > G''$) and a regime where **viscous response dominates** ($G'' > G'$). We have divided the moduli accordingly to indicate the dominant response of the material as a function of frequency. It's true that both G' and G'' are important, yet these two regimes have significance as they emerged also from video particle tracking experiments where the motion of the diffusing beads was almost "arrested" in the so-called "elastic regime" but displayed linear time dependence in the "viscous regime" (Fig. S4). We think that such a representation would make it easier for non-expert readers to understand the behavior of these condensates. **We note that this type of classification has been used in a number of reports previously in describing the behavior of Maxwell fluids**; for example Grimm et al. stated "A Maxwell fluid displays a characteristic timescale τ_M that marks the transition from a high-frequency elastic regime to a purely viscous fluid at low frequencies" (*Soft Matter*, 2011, 7, 2076), and Berret et al. stated "In contrast, the Maxwell fluid displays a crossover between a viscous and an elastic regime that occurs at a fixed value of the reduced frequency" (*Berret, J.-F. et al., Nat. Commun.* 7:10134 doi: 10.1038/ncomms10134 (2016)).

To address the reviewer's point, we have revised our language in the main text to ensure that we stress that these are viscoelastic materials and both G' and G'' are important. We have now replaced "elastic regime" with "dominant elastic response" and "viscous regime" with "dominant viscous response" in the figure legends.

3. The authors made some attempts to provide some molecular explanation for their data, but the explanation leaves much to be desired. E.g., in commenting on the much longer τ_M value of RGYGG_5 relative to the counterparts of RGRGG and RGF GG, they stated "This is consistent with the fact that Tyrosine residues are stronger drivers of phase separation as compared to Phenylalanine^{13,14}" One would think R is an even stronger driver for their peptide-RNA system. Clearly they need to come up with a consistent rationale.

We thank the reviewer for raising this point. Indeed, our results draw attention to the complex interplay of interactions (such as ionic, cation- π and π - π) that leads to emergent affinities between peptides and RNA beyond the monomer-monomer interactions that are often used to explain biophysical properties of biomolecular condensates. To provide a detailed molecular explanation of the variations in condensate LVE properties with sticker variations, we have now performed phase diagram analysis of peptide-RNA mixtures, all-atom simulations, and phase coexistence simulations. In our phase diagram analysis, we found that our peptide-RNA systems display a UCST behavior (Fig. 4b,c&h). This UCST behavior suggests that inter-chain attraction drive phase separation. Accordingly, higher attractive interactions would typically lead to a higher UCST. We observed that T_{ph} progressively increased from RGP GG to RGR GG to

RGYGG, thereby indicating increased strength of intermolecular interactions by P-to-R-to-F/Y (Fig. 4c&h). The experimental and simulation-derived temperature phase diagrams establish a direct correlation between the thermal stability of peptide-RNA condensates (attractive inter-chain interactions) and their linear viscoelastic properties (Fig. 4a,c&h). Since condensate stability is dependent on the strength of intermolecular interactions that drive phase separation, we conclude that the strength of intermolecular interactions determines the dynamical properties of peptide-RNA condensates. To test the correctness of this conclusion, we performed all-atom molecular dynamics simulations probing the interactions between a generic oligomer GXG and a trinucleotide rU3 RNA (UUU, Fig. 4d). The variable amino acid X was set to Phe, Tyr, Pro, Ser, Arg and Lys. We extracted potentials of mean force of center of mass coordinates of residues (see Methods for simulation details). Our results show a clear and robust trend of Tyr (GYG) being the stickier residue outflanking the charged Arg (GRG) and Lys (GKG, Fig. 4e&f, Fig. 9b). This is consistent with the fact that our Tyr-rich peptide [RGYGG]₅ forms condensates that have more dominant viscoelastic behavior than Arg-rich peptide and Lys-rich peptides ([RGRGG]₅ and [KGKGG]₅). Upon close inspection, we find that Tyr forms much longer lived π - π ring contacts compared to shorter-lived electrostatic and salt bridges, which also explains the higher stability of Tyr-rich condensates against salt variation (Fig. 4e). Additionally, these interactions led to [RGYGG]₅ having a higher density in the condensed phase and higher UCST than other sequences as shown by our phase coexistence simulations of full-length peptides and nucleic acids (Fig. 4g&h). We note that similar binding free energy trend (TYR>ARG>PRO) has also been obtained through restrained simulations of single solvated amino acid and nucleobases under physiological salt conditions using different force field (de Ruiter, A., & Zagrovic, B. (2015). Nucleic acids research, 43(2), 708-718). Collectively, these results confirm our understanding of the role played by intermolecular interactions in dictating the overall viscoelastic behavior of peptide-RNA condensates.

In the revised manuscript, we have described these results in a separate newly-added subsection “*The LVE properties of peptide-RNA condensates are correlated with the strength of interactions between polypeptide and RNA chains*”

Other issues:

4. Their "frequency" is likely angular frequency (ω), to be distinguished from frequency (f). The two are related by a factor 2π . ω is in units of radian per second, whereas f is in units of Hz.

We thank the reviewer for raising this point. In our case, the frequency we plotted in Hz and not in rad/s. This is usually the case for passive microrheology measurements [see Evans et al., Physical review E, 2009 <https://doi.org/10.1103/PhysRevE.80.012501>] while for active oscillatory measurements, radial frequency is often used because the calculation of the G' and G'' is done from the phase shift in the strain and the stress oscillation [see Robertson-Anderson, ACS Macro Lett. 2018, 7, 8, 968–975]. We have clearly stated that in the revised manuscript to clarify any confusion in this context among the readers.

5. The data in Fig. 2i needs to be analyzed in some theoretical model to show that the mid-point of the transition is indeed mostly determined by τ_M .

We thank the reviewer for pointing this out. After careful consideration of concerns raised by other reviewers on the utility of the flipping trap experiments, we have decided to exclude these data in the revised manuscript (including Fig. 2i of the old manuscript).

6. Eq. 12 in Supplementary Information is missing an "a" in the denominator. More seriously, the MSD data can be used to deduce the viscosity via the Stokes-Einstein relation only when the exponent alpha is 1. When alpha differs from 1, D does not even have the correct unit (i.e., $\mu\text{m}^2/\text{s}$). This may mean that the authors need to reassess all their VPT data for viscosity.

We thank the reviewer for pointing out this typographical error. We have fixed this issue. As for fitting the MSD data, we have only used this equation to fit the MSD of probe particles in [KGKGG]₅-rU40 condensates to extract the viscosity. For these condensates, the diffusivity exponent α was equal to 1 and hence the calculation of viscosity from Stokes-Einstein relation remains valid. For all other condensates (such as [RGYGG]₅-rU40 and [RGRGG]₅-rU40 which show viscoelastic behavior and nonlinear MSDs), we have not attempted to extract viscosity or diffusion coefficient values from the MSD curves. All the reported viscosities were calculated from the viscoelastic moduli obtained by pMOT using the relation $\eta = G''/\omega$.

In summary, we thank the reviewer for his/her constructive comments and suggestions. We deeply appreciate the care and effort of the reviewer in helping us to significantly improve our manuscript.

Reviewer #2 (Remarks to the Author):

Banerjee and co-workers use optical tweezers microrheology to characterize the viscoelastic properties of programmable protein-RNA condensates. The overarching claim is that they can tune the viscoelastic properties of the condensates over several orders of magnitude by varying the molecular architecture of the sticker-spacer polypeptides used in the aforementioned condensates. I find the concept intriguing and relevant to biological and engineering applications. However, there are essential controls, experiments, and details in regards to the measurement of viscoelastic properties that are needed to be able to properly evaluate the results and ensure the results are not artifacts of the experiment design. My main concerns are detailed below:

We thank the reviewer for his/her encouraging remarks. We have addressed the issues raised by the reviewer to the best of our ability and we thank him/her for the rigorous and technical revision of this manuscript. To facilitate the reviewer's assessment of our responses, we have included the key new figures both in the supplementary information of the revised manuscript as well as in this response letter.

1. The distance from the glass slide used in pOTM measurements is $\sim 2\text{-}5\ \mu\text{m}$. This is not enough distance from the surface to be able to ignore surface effects on the trapping signal. One needs to be at minimum ~ 10 bead diameters from a surface to neglect this effect. The authors show that being in the center of a condensate versus the edge does not significantly impact the results but the soft condensate interface is quite different than that of the glass slide. The authors need to demonstrate that at distances larger than $2\text{-}5\ \mu\text{m}$ from the surface their results are the same. Or they need to account for surface effects and subtract them from their signal.

This is a very important point raised by the reviewer. The effects coming from the proximity of the glass surface on the laser signal can be significant when the bead is close to the glass surface. This is especially worrisome if the laser signal is used to track the bead or calculate the forces. However, we used a bright-field camera for tracking, which is expected to eliminate many of these concerns (please see point 6 below). This is because the camera tracking does not depend on the laser signal and hence is not substantially affected by the presence of the surfaces. The function of the laser in that case is only to provide a harmonic potential for holding the bead so that long-time tracking is feasible. To address the reviewer's concerns, we have conducted a series of control experiments to probe and quantify the surface effects (or a lack thereof) on our condensates and to characterize the integrity of the optical trap. From these measurements, we find that 2-10 μm distance from the glass surface is feasible to do microrheology experiments within condensates without any spurious surface effects (we performed pMOT experiments at 2-5 μm surface-to-bead distance). **The new control experiments are described below:**

(1) We performed microrheology experiments on beads in water (not in condensates) near the glass surface. In Figure S20, we show the autocorrelation curves for a bead trapped in water at identical power and trap stiffness for various distances from the surface and tracked with the laser (not the camera). We found that if the bead is 1 μm away from the surface, the autocorrelation curves show deviations from those obtained for beads at larger distances from the surface ($>1\mu\text{m}$). This experiment was also repeated for beads in 35% (wt/vol) PEG8000 solution (tracked using the camera). We see that the autocorrelation curves are identical for the bead at 3-10 μm distance from the surface. Hence, we argue that the presence of glass surface does not affect the measured autocorrelation functions for the bead at 3-10 μm distance from the surface and therefore the obtained viscoelastic moduli, even with laser tracking. These results are included in the revised manuscript and Fig. S20.

Fig. S20. (a) A scheme illustrating the trapping of a particle in a continuous medium at variable distance h from the glass surface. (b) Normalized position autocorrelation function (NPAF) of a trapped bead at variable distance from the surface and at identical laser power within water. (c) Normalized position autocorrelation function (NPAF) of a trapped bead at variable distance from the surface and at identical laser power within a solution of 35 % wt/vol PEG8000.

(2) We performed pMOT experiments on beads trapped within [KGKGG]₅-rU40 condensates at different distances between 1 μm to 8 μm (until the upper surface of the condensate is reached). Generally speaking, the autocorrelation curves showed good agreement with each other. We find that the autocorrelation curves particularly between 2 μm to 8 μm distance from the surface are overlapping and are consistent with each other (this data was obtained using the camera tracking). This indicates that in these z-positions, the effects coming from the surface are negligible (Fig. S21a&b).

Fig. S21. Control experiments probing the feasibility of pMOT experiments within [KGKGG]₅-rU40 condensates. (a) Scheme illustrating the optical trapping of a polystyrene microsphere within peptide RNA droplet at distance h from the glass slide surface. (b) Normalized position autocorrelation function (NPAF) of a trapped microsphere within the condensate at variable distances from the glass slide surface and identical trapping power. Inset: zoomed in plot two show the variance in the NPAF values with h . (c) probability distribution of the trapped particle displacements from the center of the optical trap at variable distances from the glass slide surfaces. (d) the change in the probability distribution of particle displacements from the center of the optical trap due to increasing trapping power and trapping stiffness within peptide-RNA condensate. (e) Trap stiffness scales linearly with laser power inside peptide-RNA condensates.

(3) We calculated the distribution of the bead's displacement in x and y directions in [KGKGG]₅-rU40 condensates to explore the nature of the trapping optical potential within the droplet. The distribution showed a Gaussian dependence on the displacement, which is expected for a harmonic potential optical trap [see Neuman, K. C., & Block, S. M. (2004). *Review of scientific instruments*, 75(9), 2787-2809]. This means that the shape of the optical trapping potential is harmonic and unaltered by the presence of the glass surface in both x and y directions under these conditions. Additionally, the displacement distribution from experiments performed at variable distances from the surface showed excellent agreement (see Fig. S21c), indicating that the trap potential and the trapping stiffness are unaffected by the presence of a glass surface for our condensates at the specified distances.

(4) To further showcase the integrity of the optical trap, we measured the trap stiffness as a function of laser power inside the condensate and found a linear dependence (Fig. S21e), which is expected for optical traps up to a power of ~ 500 mW [see Sarshar, M. et al. (2014). *Journal of biomedical optics*, 19(11), 115001]. Our trapping power at 100 % laser is about 1 mW according to the manufacturer. For our experiments we used 1%-10% overall power (which at the most is ~ 100 μ W). Also, with changing laser power, we observe that the probability distribution showed narrower Gaussians as the trap stiffness was increased, further indicating that the camera tracking is valid (Fig S21d&e).

Given these experimental results, we conclude that surface effects for our particular experiments are only relevant if the bead is $\leq 1 \mu\text{m}$ away from the surface. The presented control measurements affirm this conclusion. We have performed our pMOT experiments at distances between 2-5 μm , which is above the range where surface effects can complicate data analysis. Doing the same experiments at distances $> 5 \mu\text{m}$ yielded similar results as our new experiments suggest (ACFs and bead motion is consistent across the 2-10 μm range, Fig. S21b&c). We hope these experiments and clarifications sufficiently address the reviewer's concerns on this issue. These results are discussed under "Supplementary Note 1: Effect of solid and liquid interfaces" in the revised manuscript.

Finally, we have now included results in the revised manuscript from complementary biophysical assays, including temperature-dependence of peptide-RNA phase separation (Fig. 4a-c), diffusion measurements by FRAP (Fig. S5), condensate mesh-size estimation (Figs. S6&S7), all-atom simulations (Fig. 4d-f) and phase coexistence simulations (Fig. 4g-h). **These results are described in detail in point # 2 below. These new data reaffirm our original conclusion that the inter-chain interactions between polypeptide chains and RNA govern the material properties of condensate network and that our results are not merely a manifestation of "experimental artifacts".**

2. The authors state they use carboxylated beads but there is no mention of surface passivation. Carboxylated beads that are not coated with another protein or polymer are sticky and likely interact with the protein-RNA network. This interaction will have a major impact on the results presented. For the viscoelastic properties to be accurately measured the beads can only have steric interactions with the network with no sticking. Authors need to demonstrate that the beads are not adhering to the network or need to repeat measurements with beads coated with BSA, PEG or another inert polymer (as is standard in OTM microrheology).

We thank the reviewer for this point. We agree that there is a possibility that carboxylate beads interact with the condensate network that is made of protein and RNA molecules. We have previously addressed this in a published report using [RGRGG]₅-dT40 condensates (dT40 is a single stranded DNA of the same length as rU40). In that study, we showed that the presence of the carboxylate beads did not alter the overall phase behavior of the peptide-nucleic acid system (see Fig. S3 in Alshareedah et al., Biophysical Journal 120, 1161–1169, 2021). This means that the beads are not affecting the interactions between the nucleic acids and the peptide in the dense phase significantly. Having said that, we note that some interactions may be desirable in order for beads to partition within micron-sized phase-separated condensates. To test that the beads are not affecting the interactions between the nucleic acids and the peptide significantly, we have performed a bead-halo assay on non-fluorescent carboxylate beads embedded within peptide-RNA condensates. The bead halo assay is used to probe intermolecular interactions by coating a bead with a substrate and adding the bait protein (Hayes et al., eLife 2020;9:e51685 DOI: 10.7554/eLife.51685). If the protein binds to the substrate it will coat the bead and the bead surface will have an intensity higher than the average bulk intensity. We used functionalized carboxylate beads which partition positively in peptide-RNA condensates. As our baits, we used [RGRGG]₅-Alexa594 and U10-FAM RNA to see if the beads are interacting with the peptide or the RNA inside the condensates. Our experiments show that the bead's intensity is less than or equal to the mean intensity of the probes inside the condensates, indicating no significant interactions or adsorption of the biopolymers on the bead surface (Fig. S18). This result is also consistent with a recent study that showed that the MSD of PEGylated beads and carboxylate beads is identical within

condensates formed by polyR10 ([R]₁₀) and UTP (Fisher RS & Elbaum-Garfinkle S. (2020) *Nature Communications*. Sep 15;11(1):4628). Moreover, we have previously attempted using PEGylated beads and BSA beads, only to find that they remained excluded from the condensates and hence, rheological and MSD analysis was not possible. Previous reports have indicated that PEG has repulsive interactions with polyelectrolytes and is usually excluded from complex coacervates formed by oppositely charged polymers [see (i) Marianelli, A. M. et al. (2018), *Soft Matter*, 14(3), 368-378; (ii) Park, S. et al. (2020). *Communications Chemistry*, 3(1), 1-12].

Figure S18. (a) schematic illustration of the expected outcome of the bead-halo assay in case of the presence and absence of bead-client interactions. (b) bright-field and fluorescent images of carboxylate beads within [RGRGG]₅-rU40 condensates and corresponding intensity profiles. (Top panel) The recruitment behavior of [RGRGG]₅-A594 with the carboxylate beads within peptide-RNA condensates showing that the beads do not recruit or adsorb any peptide-molecules on its surface. The intensity profile across the bead (black) shows a dip in intensity due to the absence of any [RGRGG]₅-A594 molecules on its surface. The intensity profile across a bead-free region of the droplets shows the average mean intensity. (Bottom panel) similar data but with U10-FAM RNA as the client in the bead-halo assay. These results indicate that the carboxylate beads do not interact significantly with the condensate medium as they don't recruit or concentrate the two components forming the condensates: [RGRGG]₅ and RNA.

Additionally, the findings obtained from the microrheology experiments are reaffirmed in a series of new complementary experimental assays and MD simulations. To support our original claim that peptide sequence perturbations alter the condensate dynamical properties, we performed the following additional measurements in the revised manuscript:

- (1) we performed FRAP experiments on an RNA client and found that the RNA diffusion time is significantly altered upon changing the peptide sequences from RGPGG to RGRGG to RGYGG repeat motifs. This corroborates the observed trends in the viscosity and terminal relaxation time that we report from our pMOT experiments. These results are included in the revised manuscript (Fig. S5).
- (2) we performed turbidity measurements to obtain the salt concentration required to dissolve these condensates. The presence of ions strongly affects the electrostatic interactions (by controlling the Debye screening length) that are the main driver of LLPS of oppositely charged peptide-RNA mixtures. We find orders of magnitude difference in

the salt concentration required to dissolve the condensates between [RGRGG]₅, [RGPGG]₅ and [RGYGG]₅ (Fig. S9a), again showing the exact same trend as the observed LVE properties.

- (3) we performed thermo-responsive phase separation measurements on condensates formed by [RGPGG]₅, [RGRGG]₅ and [RGYGG]₅ spanning the entire range of the LVE properties observed (see Fig 4c). As the temperature of the sample is increased, LLPS was diminished for these peptide-RNA condensates revealing an Upper Critical Solution Temperature behavior (UCST). This UCST behavior suggests that inter-chain attraction drives phase separation. Accordingly, higher attractive interactions would typically lead to a higher UCST. We observed that T_{ph} progressively increased from RGPGG to RGRGG to RGYGG repeat motifs, thereby indicating increased strength of intermolecular interactions by P-to-R-to-F/Y substitution (Fig. 4c). This temperature phase diagram establishes a direct correlation between the thermal stability of peptide-RNA condensates (attractive inter-chain interactions) and their linear viscoelastic properties.
- (4) We estimated the difference in the condensate network mesh size (the pore size) between the most dynamic ([RGPGG]₅) and least dynamic condensates ([RGYGG]₅) using fluorescently-labeled dextran molecules of various sizes (See point 3 below). Our findings suggest that the mesh size of [RGYGG]₅-rU40 condensates is smaller than [RGPGG]₅-rU40 condensates. This is consistent with the observed difference in the LVE behavior of these condensates (mesh sizes were in the order of 10 nm, Figs. S6&S7).
- (5) we performed all-atom molecular dynamics simulations probing the interactions between a generic oligomer GXG and a trinucleotide rU3 RNA (UUU, Fig. 4d). The variable amino acid X was set to Phe, Tyr, Pro, Ser, Arg and Lys. Results show a clear and robust trend of Tyr (GYG) being the sticker residue outflanking the charged Arg (GRG) and Lys (GKG, Fig. 4e&f; Fig. S9b). These differences in interactions were manifested in higher UCST and higher density within the condensed phase as per phase coexistence simulations that we conducted with full-length peptides and nucleic acids (Fig. 4g&h). This is consistent with the fact that our Tyr-rich peptide [RGYGG]₅ forms condensates that have more dominant viscoelastic behavior than Arg-rich peptide and Lys-rich peptides ([RGRGG]₅ and [KGKGG]₅). Upon close inspection of the free energy profiles from the all-atom simulations, we find that Tyr forms much longer lived π - π ring contacts compared to shorter-lived electrostatic and salt bridges, which also explains the higher stability of Tyr-rich condensates against salt variation (Fig. 4e).

Given that these additional experiments were done without any carboxylate modified beads, yet the results are consistent with our LVE properties obtained from pMOT and support our model that stronger inter-chain interactions determine the dynamical behavior of condensates, we conclude that the LVE properties variance between the studied sequences is solely dictated by the protein sequence, and the bead-condensate interaction is not significant in this context.

3. Another issue that is not addressed is the small bead size (1 μ m diameter?). How does the bead size compare to the mesh or pore size of the condensate network? It seems that the varying viscoelastic properties measured could easily be a result of changing network architecture that changes the pore size. If the bead is not sufficiently larger than the pore size of all the condensates tested then the measured effects can not be considered to represent the viscoelasticity of the condensate. Authors need to show evidence of or calculations of the pore size and show that the bead size is substantially larger than that. Alternatively, they could

perform measurements using varying bead sizes (larger) and show that the results are not affected by changing bead sizes.

We thank the reviewer for raising this point. The mesh size of protein-RNA condensates is within the order of nanometers (typically < 10 nm). This has been previously reported for multiple types of condensates [see Wei, M. T. et al. (2017). Nature chemistry, 9(11), 1118.] and for complex coacervates (see Spruijt, E. et al. (2013). Macromolecules, 46(11), 4596-4605.). We have previously used dextran polymers to inspect the permeability of condensates formed by cationic R-rich polypeptides and poly(U) RNA [see Alshareedah, I. et al. (2020). Proceedings of the National Academy of Sciences, 117(27), 15650-15658.] and found that such condensates have a mesh size of ~ 2.3 nm. To confirm this, we have now performed the same permeability assay by using dextran polymers of varying molecular weights and hydrodynamic radii and inspected their diffusion through condensates formed by [RGYGG]₅ and rU40 (the most viscoelastic and least dynamic) and condensates formed by [RGPGG]₅-rU40 (the most dynamic) and found that the mesh size is ~ 4.5 -6 nm (Figs S6&S7). Therefore, the use of 1 μ m beads is suitable as it is larger than the pore size of these condensates by three orders of magnitude.

Fig. S6. Mesh size-determination experiment using TMR labeled Dextran molecules with variable molecular weights for [RGPGG]₅-rU40 condensates. Upper panel shows the

condensates as visualized by Alexa488-tagged [RGPGG]₅ (1% labeling ratio). Middle panel shows the partition behavior of Dextran molecules within peptide-RNA condensates. Lower panel shows corresponding intensity profiles for Dextran molecules. The numbers in brackets indicate the estimated hydrodynamic radius of the Dextran molecules in aqueous solutions.

4. All of the viscoelastic data presented is very (suspiciously) clean. Yet there is no mention of smoothing. Is this really the raw data presented without smoothing? It seems that with only 9-15 trials and with biological samples there would be more noise in the signal. Especially, in Fig S9 where individual trials are plotted. Please comment.

In our initial submission, we had clearly mentioned in the method section of the SI that the autocorrelation curves were fitted with a multi-exponential function before performing the Fourier transform to reduce the noise that comes from the numerical artifacts. To make this point clearer, an example of the raw and fitted autocorrelation functions was shown in the original manuscript (Fig. 1d). This analysis was done following the same protocol as Tassieri *et al.* [Tassieri, M. Microrheology with Optical Tweezers: Principles and Applications. CRC Press, (2016); Preece, D. et al. Journal of optics 13, 044022 (2011); Tassieri, M., Evans, R., Warren, R. L., Bailey, N. J. & Cooper, J. M. New Journal of Physics 14, 115032 (2012); Tassieri, M. et al. Physical Review E 81, 026308 (2010)].

To make it clearer and remove any confusion in the revised manuscript, we have now included a Supplementary Note (SI Note-1) that describes the analysis procedure in full detail. The analysis is summarized in a new SI figure (Fig. S15).

Fig. S15. Flow chart showing pMOT data analysis procedure to obtain the viscoelastic moduli of phase-separated condensates. (a-e) various steps of the procedure are described in the text of SI Note-1.

Briefly, we calculated raw autocorrelation functions from the experimental trajectories. Fitting was done to the autocorrelation functions using a multi-exponential function (see Equation 11 in Supplementary Note-1). Next, Fourier transforms of the fitted autocorrelation functions were calculated and substituted into Equation 9 (SI-Note 1) to obtain the viscoelastic moduli. For statistics, we have now added error bars to the fitted moduli by averaging the individual moduli from different condensates (see Fig. S15). These statistics and noise were previously accounted for in reporting the viscosity and terminal relaxation time values (see Fig. 2e&f). For the cleanliness of the data, **we have used the fits to smooth out the noise coming from autocorrelation values at long timescale. The autocorrelation is a statistical function and therefore cannot be exact except at infinite measurement time, which is practically unrealizable. Therefore, several smoothing procedures have been described in the literature such as fitting or cubic spline interpolation (see [Tassieri, M. Microrheology with Optical Tweezers: Principles and Applications. CRC Press, (2016); Preece, D. et al. Journal of optics 13, 044022 (2011); Tassieri, M., Evans, R., Warren, R. L., Bailey, N. J. & Cooper, J. M. New Journal of Physics 14, 115032 (2012); Tassieri, M. et al. Physical Review E 81, 026308 (2010)]).** We followed the steps described in these references.

In order to improve the presentation of the data, we have calculated the average moduli for each peptide-RNA system from all the pMOT trials as well as the standard deviation. We have replotted the moduli in the paper with error bars (see Fig 2b in the revised manuscript as an example). The error is estimated from the smoothed moduli of several trials ($n \geq 12$). We also have included a comparison between G' and G'' calculated from fitted autocorrelation functions and raw autocorrelation functions for all the peptide-RNA systems in our study in the SI (see Figs. S24&S25) in order for the readers to compare. Lastly, we added two figures showing the fits of the autocorrelation functions to all the systems tested in this work (Figs. S22&23). We thank the reviewer for drawing our attention to these issues and hope these revisions and clarifications are sufficient to address the same. Two of these figures are shown below.

Fig. S22. Representative plots of the calculated normalized position autocorrelation function (NPAF) and the multi exponential fit (equation-11) for various peptide-RNA condensate systems shown in Fig 1-3 in the main text.

Fig. S25. Example plots of the viscoelastic moduli calculated from raw (blue and green) and fitted (black and red) autocorrelation functions (NPAF) of various peptide-NA condensate systems shown in Figs 1-2 in the main text.

5. Authors should show fits of the autocorrelation functions with multi-exponential functions (described in SI) to assess goodness of fit. I suspect that the very clean data I discussed in #4 is a result of this fitting but its not clear. It's not clear what the actual data would look like without relying on the fit. This delineation needs to be made clear

We thank the reviewer for raising this point, we have included the autocorrelation curves and the fits as well as the smoothed and raw moduli in the SI for each peptide-RNA system tested in this study (see Figs. S22-25). Please also refer to our response to point 4 above in this context.

6. In pOTM experiments authors use video particle tracking to measure thermal fluctuations. This method is inherently lower accuracy than using the laser signal. Why was this chosen? And what is the centroid positioning accuracy? This data needs to be shown and uncertainty

provided.

The purpose for using the camera is to avoid scattering issues of the laser from its path towards the detector. The two-phase interface above the bead might refract the laser light and give erroneous position detection especially if it is a curved interface (see the diagram below). This problem is not apparent when probing bulk samples such as water or glycerol where there is no curved interface between the bead and the detector. The use of a camera, although low resolution as pointed out by the reviewer, eliminates these concerns. Therefore, although it's true that laser detection is superior in terms of spatiotemporal resolution, many factors can contribute to the inaccuracies in laser detection data that are related to the sample conditions and the presence of interfaces, making it inconvenient to use for our current experimental systems such as beads within micron-sized phase separated droplets.

Figure. A scheme showing the laser refractions within a protein-RNA condensate.

In addition, laser tracking is possible only when the laser is a priori calibrated. This requires the use of a camera for calibration since the viscosity of the condensate is unknown and therefore calibration with power spectral analysis is more challenging. Other studies that used laser tracking to measure the LVE properties of droplets calibrated the laser in water and then performed active OT experiments (Jawerth LM, et al. (2018) Physical review letters 121(25):258101). However, since we don't have a priori knowledge of the viscosity of the condensate, the laser cannot be used for tracking unless it is calibrated, which can be done using the equipartition theorem and camera-based position detection (S Keen et al 2007 J. Opt. A: Pure Appl. Opt. 9 S264). Hendricks and Goldman have discussed this issue nicely in a paper that focuses on the challenges of calibrating optical tweezers in cells [Hendricks, A. G., & Goldman, Y. E. (2017). Measuring molecular forces using calibrated optical tweezers in living cells. In Optical Tweezers (pp. 537-552). Humana Press, New York, NY]. Due to these complexities, we opted to use the camera for both calibration and tracking in this current study.

The centroid position frequency for our study is ~ 1.5 nm, we measured it by trapping beads in water at different laser powers and observed the minimum standard deviation in the positional fluctuations of the bead around the center of the trap (**Fig. S16 in the revised manuscript**).

Fig. S16. (a) Trajectories of a bead trapped in water at different laser trapping powers (100% laser is ~ 1 mW). **(b)** Displacement distribution of the bead from the center of the trap. Data is shown for trapping power of 5%. σ is the standard deviation of the positional fluctuations in the X or Y direction. **(c)** The standard deviation ($\sigma = \sqrt{\text{variance}}$) of the trapped bead position fluctuations under different laser trapping power values. The values of σ saturate at 1.5 nm, which is taken as the centroid positioning accuracy.

7. Related to #6 why is the laser signal used for the trap flipping experiments and video tracking used for pOTM measurements? What is the DAQ rate for the laser signal?

The DAQ rate of our laser is 72.8 kHz. We used laser signal for trap flipping experiments because it has a higher time resolution than the camera and therefore it was possible to detect the dead time of the experiment (the time taken by the trap to shift its position). Furthermore, as we are shifting the trap by 300 nm, the bead travels appreciable amount of distance in one direction, which can be tracked easily by the laser. Since no material properties (G' , G'') were extracted from these measurements, the use of laser tracking was deemed appropriate. **However, these considerations are not relevant anymore since following the reviewer's recommendation below, we have decided to omit the flipping trap experimental data from the revised manuscript.**

8. I'm struggling to see the point of the 'step-strain' flipping experiment. The presented timescales do not appear to correlate with any physically relevant relaxation mechanism or timescale and do not seem to provide any additional information. I also do not see how this measurement is an 'independent verification' of results? This measurement, like pOTM, also depends on the size of bead relative to the condensate pore size. It also depends on the distance the stage is moved (which is not justified in the paper). If the authors want to use this type of analysis it needs to be unpacked more and these issues need to be addressed.

We agree with the reviewer that the step-strain experiment was used in our original manuscript has no significant new information are obtained from these measurements. Since we have included results from several complementary assays (phase diagram analysis, diffusion measurements by FRAP, condensate mesh size determination), all-atom simulations, and phase coexistence simulations to provide a detailed molecular picture, we have decided to omit these data from the revised manuscript.

9. The closeness of their measured water viscosity to reported values is not great (>20% difference). This makes me question their methods. One reason for this discrepancy could be surface affects (being too close to the glass slide). Authors should check this.

We thank the reviewer for raising this point. We argue that the variance of measured water viscosity from the literature report is not as significant as the reviewer is suggesting due to the following reasons: our experimental method relies on calculating accurate autocorrelation functions from the motion of beads within a medium. Water is a very low viscosity material compared to condensates that we have studied and hence the motion of the particle in water is very fast, resulting in its autocorrelation dropping to zero at timescales of a few milliseconds. Hence the number of points in the water autocorrelation function is low, which can cause higher uncertainties in water viscosity measurements. The value of the uncertainty in water viscosity is 0.0002 Pa.s, which is much lower than all the viscosity values of the condensate materials tested in this study. For water, the fact that it has a very low viscosity as compared to condensates makes the error percentage large, but with high viscosity materials such as the peptide-RNA condensates, the percentage error remains reasonable. **An improvement to the measurement of water viscosity can be done by using a larger bead, which will have slower motion and longer lived autocorrelation. To test this idea, we have used 2 μm beads to measure water viscosity and found it to be 0.93 ± 0.03 mPa.s, which is closer to the literature report.** This new data is now reported in Figure S17.

Fig. S17. (a) Viscoelastic moduli of water using the pMOT assay with 2 μm polystyrene beads. **(b)** The viscosity of water as measured from the pMOT assay (water viscosity is 0.91 mPa.s).

Additionally, we would like to point out that the pMOT technique used in our study has been previously developed and successfully used for several types of soft materials (see for example [Tassieri, M. Microrheology with Optical Tweezers: Principles and Applications. CRC Press, (2016); Preece, D. et al. Journal of optics 13, 044022 (2011); Tassieri, M., Evans, R., Warren, R. L., Bailey, N. J. & Cooper, J. M. New Journal of Physics 14, 115032 (2012); Tassieri, M. et al. Physical Review E 81, 026308 (2010)]). Our current experimental setup, as with any other experimental techniques in general, provides accurate results for materials within a range of viscoelasticity values, and materials above or below that range may need improved detection efficiencies (such as faster camera tracking). As for the concerns regarding the effects of glass surface we have performed the water measurements of viscosity >20 μm away from any glass surfaces, please refer to our response in Point # 1 above and SI Note-1 in the revised version of

the manuscript for controls regarding the effect of surfaces on pMOT experiments in water and in condensates.

Reviewer #3 (Remarks to the Author):

This is an interesting and very important study that adds significantly to the field and will have a noticeable impact. The manuscript is well-written and concise. Reported data are of utmost importance and definitely will be of great interest to many researchers. The authors conclude this exceptional study stating "the application of laser tweezer-based microrheology will pave the path for understanding the molecular origins of viscoelasticity in membrane-less organelles". I could not agree more with this assertion.

Vladimir N. Uversky

We thank the reviewer for his remarks and appreciate his encouragement.

Reviewers' Comments:

Reviewer #1:

Remarks to the Author:

This revision has incorporated additional data that addressed major gaps identified in the previous reviews. I very much appreciate the authors' efforts in attempting to reaching a molecular explanation for their data on viscoelasticity. Still, a few concerns remain.

1. The authors made some improvements regarding the use of simplistic language such as a viscous vs. an elastic regime, but such practice continues. In particular, they call some condensate least "dynamic" and another most "dynamic". This criterion seems to be based on the terminal relaxation time (τ_M) or the zero-shear viscosity, or both: long τ_M (and/or high viscosity) = less dynamic. This labeling is unhelpful and indeed misleading.

2. Regarding MSD, the reply stated they only extracted viscosity from condensates with $\alpha=1$. However, the text around eq (1) in Methods still expresses MSD as $4D\tau^\alpha + b$, and called this D as "diffusion coefficient", which was then converted to viscosity via eq (1). Again, my point is that when α is not 1, the units of D are not even those expected of a real diffusion coefficient and eq (1) would make no sense.

3. The authors have now done more control experiments in support of their methodology, where the trap stiffness was determined by the variance of the position as reported by a camera. It is important to verify this calibration. One way to verify is to do calibration in a solvent with known viscosity, by acquiring the power spectrum of a trapped bead. The corner frequency of the power spectrum, along with the known viscosity and bead radius, allows for the calibration of the trap stiffness.

4. They now report the trap stiffnesses for one condensate in Fig. S21. Stiffness data should be reported for all the condensates studied, and for water as control.

Reviewer #2:

Remarks to the Author:

The authors have performed new measurements and analyses to thoroughly address my questions and concerns. I recommend for publication.

Point-by-point response to Reviewers' comments

Reviewers' comments are colored in **black**, the authors' response is colored in **blue**. Hyperlinked references that are cited in our responses are underlined. The major changes made are highlighted in **yellow** in the revised manuscript and listed in the context of individual comments of the reviewers below. Figures or figure panels reporting new data are included in this response for clarity and ease of access, figure captions are colored in **dark red**.

Reviewer #1 (Remarks to the Author):

This revision has incorporated additional data that addressed major gaps identified in the previous reviews. I very much appreciate the authors' efforts in attempting to reaching a molecular explanation for their data on viscoelasticity. Still, a few concerns remain.

We thank the reviewer for his/her appreciation. Below, we address the remaining concerns of the reviewer to the best of our ability.

1. The authors made some improvements regarding the use of simplistic language such as a viscous vs. an elastic regime, but such practice continues. In particular, they call some condensate least "dynamic" and another most "dynamic". This criterion seems to be based on the terminal relaxation time (τ_M) or the zero-shear viscosity, or both: long τ_M (and/or high viscosity) = less dynamic. This labeling is unhelpful and indeed misleading.

As per this suggestion, we have omitted the use of the terms *least* and *most dynamic*. We replaced "*least dynamic*" with "*strongest viscoelastic behavior*" and "*most dynamic*" with "*weakest viscoelastic behavior*" to say that both G' and G'' are the largest or the smallest, respectively. We feel that this revised terminology is appropriate for describing these condensates in a comparative manner.

2. Regarding MSD, the reply stated they only extracted viscosity from condensates with $\alpha=1$. However, the text around eq (1) in Methods still expresses MSD as $4D\tau^\alpha + b$, and called this D as "diffusion coefficient", which was then converted to viscosity via eq (1). Again, my point is that when α is not 1, the units of D are not even those expected of a real diffusion coefficient and eq (1) would make no sense.

We have addressed this point in the revised manuscript. We have rewritten the MSD part to eliminate the apparent contradiction between D and the exponent α , and fixed Equation-1 (see below). The revised section now reads (SI; page-3):

"...In general, the MSD scales as a power law with time ($MSD \propto t^\alpha$), where α is diffusivity exponent. In the special case when $\alpha = 1$ (as in [KGKGG]₅-rU40 condensates), the MSD can be fitted to extract the diffusion coefficient ($MSD = 4D\tau + b$). In this equation, D is the diffusion coefficient and b is a constant accounting for the noise. The condensate viscosity η was then calculated from the Stokes-Einstein equation

$$\eta = \frac{k_B T}{6\pi D a} \quad (1)$$

Where k_B is Boltzmann constant, T is the temperature, and a is the particle radius."

3. The authors have now done more control experiments in support of their methodology, where the trap stiffness was determined by the variance of the position as reported by a camera. It is important to verify this calibration. One way to verify is to do calibration in a solvent with known viscosity, by acquiring the power spectrum of a trapped bead. The corner frequency of the power spectrum, along with the known viscosity and bead radius, allows for the calibration of the trap stiffness.

To verify the camera-based thermal calibration method utilizing the equipartition theorem, we also independently performed calibration where we trapped a microsphere in water and tracked it with the trapping laser (through a Quadrant photodiode). Subsequently, we performed the power spectral density (PSD) analysis on the data collected from the optical trap and extracted the trap stiffness. Simultaneously, we extracted the trap stiffness from the camera-based trajectories through the equipartition theorem (EPT) using the same bead under identical conditions. Both calibration methods gave consistent trap stiffness values (see **Fig. S21 f&g**). We hope this addresses the reviewer's question.

Additionally, we note that the trap calibration with equipartition method is well established and has been used in numerous studies that precede the present work (please see [Jun, Y. et al., (2014). *Biophysical journal*, 107(6), 1474-1484.], [Sarshar, M., Wong, W., & Anvari, B. (2014). *Journal of biomedical optics*, 19(11), 115001], [Neuman, K. C., & Block, S. M. (2004). *Review of scientific instruments*, 75(9), 2787-2809.] and [Gieseler, J. et al., (2021). *Advances in Optics and Photonics*, 13(1), 74-241] as well as references within this manuscript).

Fig. S21. (f) A representative power spectrum of the fluctuations of a 4.6 μm bead diffusing in water and held by an optical trap. These fluctuations are tracked through the optical trap using a quadrant photodiode (Lumicks, C-trap). The fit shown here is a Lorentzian fit ($PSD = a/(f^2 + f_c^2)$) that gives the corner frequency f_c from which the trap stiffness is calculated given the viscosity of the medium (water) is known. The PSD calibration is done through a built-in routine provided with

the optical trap instrument (Lumicks, C-trap) following previously established methods²⁶. (g) Comparison between laser-based calibration via power spectral density analysis (PSD) and the camera-based calibration via the equipartition theorem (EPT) used in this study for the same bead in (g) over multiple calibration trials.

4. They now report the trap stiffnesses for one condensate in Fig. S21. Stiffness data should be reported for all the condensates studied, and for water as control.

We have plotted the trap stiffness values for all of our condensates in a new SI Figure in the revised manuscript (Fig. S26).

Fig. S26. Values of the trap stiffness inside the condensate for all the condensates examined in this work as well as water. Error bars represent the range of the data.

Reviewer #2 (Remarks to the Author):

The authors have performed new measurements and analyses to thoroughly address my questions and concerns. I recommend for publication.

We are glad to learn that our new data satisfied the reviewer's questions and we appreciate his/her rigorous work in reviewing this manuscript.